# UFO² : The Desktop AgentOS

**Chaoyun Zhang[1]**, **He Huang[1]**, **Chiming Ni[2]**, **Jian Mu[3]**, **Si Qin[1]**, **Shilin He[1]**, **Lu Wang[1]**, **Fangkai Yang[1]**, **Pu Zhao[1]**, **Bo Qiao[1]**, **Chao Du[1]**, **Liqun Li[1]**, **Yu Kang[1]**, **Zhao Jiang[1]**, **Suzhen Zheng[1]**, **Rujia Wang[1]**, **Jiaxu Qian[4]**, **Minghua Ma[1]**, **Jian-Guang Lou[1]**, **Qingwei Lin[1]**, **Saravan Rajmohan[1]**, **Dongmei Zhang[1]**
[1]*Microsoft*    [2]*ZJU-UIUC Institute*    [3]*Nanjing University*    [4]*Peking University*

**Reviewed on OpenReview:** *https://openreview.net/forum?id=iAuZVWCduc*

## Abstract

Recent *Computer-Using Agents* (CUAs), powered by multimodal large language models (LLMs), offer a promising direction for automating complex desktop workflows through natural language. However, most existing CUAs remain conceptual prototypes, hindered by shallow OS integration, fragile screenshot-based interaction, and disruptive execution.

We present **UFO²**, a multiagent *AgentOS* for Windows desktops that elevates CUAs into practical, system-level automation. UFO² features a centralized HOSTAGENT for task decomposition and coordination, alongside a collection of application-specialized APPAGENTs equipped with native APIs, domain-specific knowledge, and a unified GUI–API action layer. This architecture enables robust task execution while preserving modularity and extensibility. A hybrid control detection pipeline fuses Windows UI Automation (UIA) with vision-based parsing to support diverse interface styles. Runtime efficiency is further enhanced through speculative multi-action planning, reducing per-step LLM overhead. Finally, a *Picture-in-Picture* (PiP) interface enables automation within an isolated virtual desktop, allowing agents and users to operate concurrently without interference.

We evaluate UFO² across over 20 real-world Windows applications, demonstrating substantial improvements in robustness and execution accuracy over prior CUAs. Our results show that deep OS integration unlocks a scalable path toward reliable, user-aligned desktop automation.

The source code of UFO² is publicly available at https://github.com/microsoft/UFO/, with comprehensive documentation provided at https://microsoft.github.io/UFO/.

## 1 Introduction

Automation of desktop applications has long been central to improving workforce productivity. Commercial Robotic Process Automation (RPA) platforms such as UiPath UiPath (2025), Automation Anywhere Anywhere (2025), and Microsoft Power Automate Microsoft (2025) exemplify this trend, using predefined scripts to replicate repetitive user interactions through the graphical user interface (GUI) Hofmann et al. (2020); Madakam et al. (2019). However, these script-based approaches often prove fragile in dynamic, continuously evolving environments Pramod (2022). Minor interface changes can break the underlying automation scripts, requiring manual updates and extensive maintenance effort. As software ecosystems grow increasingly complex and heterogeneous, the brittleness of script-based automation severely limits scalability, adaptability, and practicality.

Recently, *Computer-Using Agents* (CUAs) Zhang et al. (2024a) have emerged as a promising alternative. These systems leverage advanced multimodal large language models (LLMs) Zhao et al. (2023); Zhang et al. (2024d) to interpret diverse user instructions, perceive GUI interfaces, and generate adaptive actions

---

*Chaoyun Zhang is the corresponding author: `vyokky@163.com`

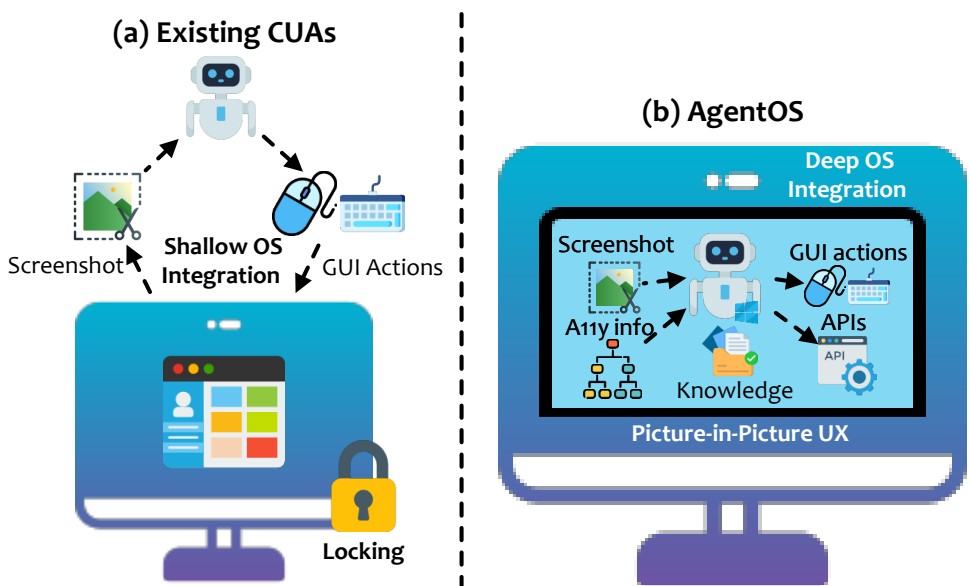

Figure 1: A comparison of (a) existing CUAs and (b) desktop AgentOS UFO$^2$.

(*e.g.*, mouse clicks, keyboard inputs) without fixed scripting Zhang et al. (2024a). Early prototypes such as UFO Zhang et al. (2024b), Anthropic Claude Anthropic (2024), and OpenAI Operator OpenAI (2025) demonstrate that such LLM-driven agents can robustly automate tasks too complex or ambiguous for conventional RPA pipelines. Yet, despite these advances, current CUA implementations remain largely *conceptual*: they mainly optimize visual grounding or language reasoning Wang et al. (2024b); Qin et al. (2025); OpenAI (2025); Zheng et al. (2025); Niu et al. (2024), but give little attention to *system-level* integration with desktop operating systems (OSs) and application internals (Figure 1 (a)).

Reliance on raw GUI screenshots and simulated input events has several drawbacks. First, visual-only inputs can be noisy and redundant, increasing cognitive overhead for LLMs and reducing execution efficiency Zhang et al. (2025). Second, existing CUAs rarely leverage the OS's native accessibility interfaces, application-level APIs, or detailed process states—a missed opportunity to significantly enhance decision accuracy, reduce latency, and enable more reliable execution. Finally, simulating mouse and keyboard events on the primary user desktop locks users out during automation, imposing a poor user experience (UX). These limitations must be resolved before CUAs can mature from intriguing prototypes into robust, scalable solutions for real-world desktop automation, and motivates our central research question:

> *How can we build a robust, deeply integrated system for desktop automation that flexibly adapts to evolving interfaces, reliably orchestrates diverse applications, and minimizes disruption to user workflows?*

In response, we present **UFO$^2$**, a new *AgentOS* for Windows that reimagines desktop automation as a first-class operating system abstraction. Unlike prior CUAs that treat automation as a layer atop screenshots and simulated input events, UFO$^2$ is architected as a deeply integrated, multiagent execution environment—embedding OS capabilities, application-specific introspection, and domain-aware planning into the core automation loop, as illustrated in Figure 1(b).

At its foundation, UFO$^2$ provides a modular, system-level substrate for natural language-driven automation. A centralized coordinator, the HOSTAGENT, interprets user instructions, decomposes them into semantically meaningful subtasks, and dynamically dispatches execution to specialized APPAGENTS —expert modules tailored for specific Windows applications. Each APPAGENT is equipped with an extensible toolbox of application-specific APIs, a hybrid GUI–API action interface, and integrated knowledge about the applica-

tion's capabilities and semantics. This architecture enables robust orchestration across multiple concurrent applications, supporting workflows that span Excel, Outlook, Edge, and beyond.

To enable reliable execution across the full spectrum of application UIs, UFO$^2$ introduces a hybrid control detection pipeline, combining Windows UI Automation (UIA) APIs with advanced visual grounding models Lu et al. (2024). This allows agents to introspect and act on both standard and custom UI components, bridging the gap between structured accessibility trees and pixel-level perception. Moreover, UFO$^2$ continuously incorporates external documentation, patch notes, and past execution traces into a unified vectorized memory layer, enabling each APPAGENT to incrementally refine its behavior without retraining.

At the interaction layer, UFO$^2$ exposes a unified GUI–API execution model, where agents seamlessly combine traditional GUI actions (*e.g.*, clicks, keystrokes) with native Windows or application-specific APIs via configurable Model Context Protocol (MCP) servers Hou et al. (2025). This hybrid approach improves execution efficiency, reduces brittleness to UI layout changes, and enables more expressive, higher-level operations. To further minimize the latency associated with LLM-based action planning, UFO$^2$ incorporates a speculative multi-action execution engine that proactively infers and validates action sequences using lightweight control-state checks at a single inference step—substantially reducing inference overhead without compromising correctness.

Finally, to ensure a practical and non-intrusive user experience, UFO$^2$ introduces a novel Picture-in-Picture (PiP) interface: a secure, nested desktop environment where agents can execute independently of the user's main session. Built atop Windows' native remote desktop loopback infrastructure, PiP enables seamless, side-by-side user–agent multitasking without disruption on the user's main desktop, addressing one of the most persistent UX limitations of existing CUAs.

Together, these design principles position UFO$^2$ not just as a smarter agent, but as a new OS-level abstraction for automation—transforming desktop workflows into programmable, composable, and robust entities. In summary, this paper makes the following contributions:

- **Deep OS Integration:** We design and implement UFO$^2$, a multiagent AgentOS that deeply embeds within the Windows OS, orchestrating desktop applications through introspection, API access, and fine-grained execution control.

- **Unified GUI–API Action Layer:** We propose a hybrid action interface that bridges traditional GUI interactions with application-native API calls via configurable MCP servers, enabling flexible, efficient, and robust automation.

- **Hybrid Control Detection:** We introduce a fusion pipeline combining UIA metadata with vision-based detection to achieve reliable control grounding even in non-standard interfaces.

- **Continuous Knowledge Integration:** We build a retrieval-augmented memory that integrates documentation and historical execution logs, allowing agents to improve autonomously over time without retraining.

- **Speculative Multi-Action Execution:** We reduce LLM invocation overhead by predicting and validating action sequences ahead of time using UI state signals.

- **Non-Disruptive UX:** We develop a nested virtual desktop environment that allows automation to proceed in parallel with user activity, avoiding interference and improving adoptability.

- **Comprehensive Evaluation:** We evaluate UFO$^2$ across 20+ real-world Windows applications, showing consistent improvements in success rate, execution efficiency, and usability over state-of-the-art CUAs like Operator.

Overall, UFO$^2$ advances the vision of OS-native automation by shifting the paradigm from GUI scripting to structured, programmable application control. Even when paired with general-purpose models like GPT-4o, UFO$^2$ outperforms dedicated CUAs by over 10%, highlighting the transformative impact of system-level integration and architectural design.

## 2 Background

### 2.1 The Fragility of Traditional Desktop Automation

For decades, desktop automation has relied on brittle techniques to replicate human interactions with GUI-based applications. Commercial RPA (Robotic Process Automation) tools—such as UiPath UiPath (2025), Automation Anywhere Anywhere (2025), and Microsoft Power Automate Microsoft (2025)—operate by recording and replaying mouse movements, keystrokes, or rule-based scripts. These systems rely heavily on surface-level GUI cues (*e.g.*, pixel regions, window titles), offering little introspection into application state.

While widely deployed in enterprise settings, traditional RPA systems exhibit poor robustness and scalability Siderska et al. (2023). Even minor UI updates—such as reordering buttons or relabeling menus—can silently break automation scripts. Maintaining correctness requires frequent human intervention. Furthermore, these tools lack semantic understanding of application workflows and cannot reason about or adapt to novel tasks. As a result, RPA tools remain constrained to narrow, repetitive workflows in stable environments, far from general-purpose automation.

### 2.2 Rise of Computer-Using Agents

Recent advances in large language models (LLMs) and multimodal perception have enabled a new class of automation systems, referred to as *Computer-Using Agents* (CUAs) Zhang et al. (2024a); Zhao et al. (2023); Zhang et al. (2024d). CUAs aim to generalize across applications and tasks by leveraging LLMs to interpret user instructions, perceive GUI layouts, and synthesize actions such as clicks and keystrokes. Early CUAs like UFO Zhang et al. (2024b) demonstrated that multimodal models (*e.g.*, GPT-4V Yang et al. (2023b)) could map natural language requests to sequences of GUI actions with no hand-crafted scripts. More recent industry prototypes, including Claude-3.5 (Computer Use) Anthropic (2024) and OpenAI Operator OpenAI (2025), have pushed the envelope further, performing realistic desktop workflows across multiple applications.

These CUAs represent a promising evolution from static RPA scripts to adaptive, general-purpose automation. However, despite their sophistication, current CUAs largely remain research prototypes, constrained by architectural and systems-level limitations that impede real-world deployment.

### 2.3 Systems Challenges in CUAs

Current CUAs fall short in three fundamental ways, which we argue stem from missing operating system abstractions:

**(1) Lack of OS-Level Integration.** Most CUAs interact with the system through screenshots and low-level input emulation (mouse and keyboard events). They ignore rich system interfaces such as accessibility APIs, application process state, and native inter-process communication mechanisms (*e.g.*, shell commands, COM interfaces Gray et al. (1998)). This superficial interaction model limits reliability and efficiency—every action must be inferred from pixels rather than structured state.

**(2) Absence of Application Introspection.** CUAs typically operate as generalists with limited awareness of application-specific capabilities. They treat all interfaces uniformly, lacking the ability to leverage built-in APIs or vendor documentation. As a result, they cannot reason over high-level concepts unless such flows are explicitly embedded in the model. This rigidity limits their generalization and makes maintenance expensive.

**(3) Disruptive and Unsafe Execution Model.** Most CUAs drive automation directly on the user's desktop session, hijacking the real mouse and keyboard. This design prevents users from interacting with their system during execution, introduces interference risk, and violates isolation principles fundamental to safe system design. Long-running tasks—especially those involving multiple LLM queries—can monopolize the session for minutes at a time.

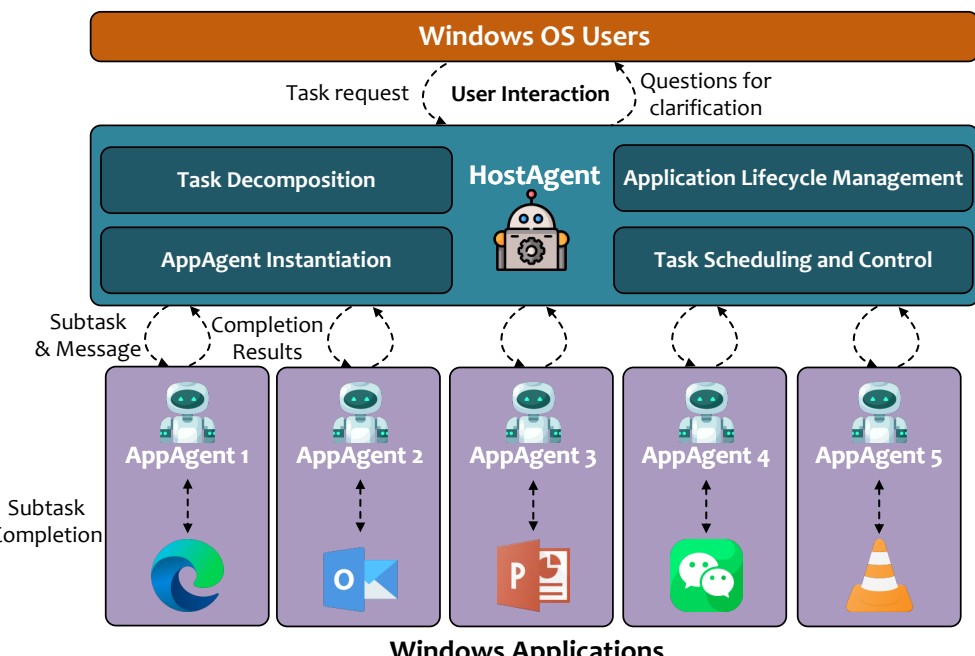

Figure 2: An overview of the architecture of $UFO^2$.

## 2.4 Missing Abstraction: OS Support for Automation

Despite growing demand for intelligent, language-driven automation, existing operating systems offer no first-class abstraction for exposing GUI application control to external agents. In contrast to system calls, files, or sockets, GUI workflows remain opaque and non-programmable. As a result, both RPA and CUA systems are forced to operate as ad-hoc layers atop the GUI, with no unified substrate for execution, coordination, or introspection.

This paper argues that automation should be elevated to a system primitive. We present **$UFO^2$**, a new *AgentOS* for Windows that addresses these limitations by embedding automation as a deeply integrated OS abstraction—exposing GUI controls, application APIs, and task orchestration as programmable, inspectable, and composable system services.

## 3 System Design of $UFO^2$

Motivated by the challenges highlighted in Section 2, $UFO^2$ is designed to seamlessly interpret natural language user requests and reliably automate tasks across a wide range of Windows applications. This section provides an architectural overview of $UFO^2$ (Section 3.1) and explains how each component is deeply integrated with the underlying OS to overcomes the pitfalls of current CUAs, ultimately enabling a practical, robust AgentOS for desktop automation.

### 3.1 $UFO^2$ as a System Substrate for Automation

Figure 2 presents the high-level architecture of $UFO^2$, which provides a structured runtime environment for task-oriented automation on Windows desktops. Deployed as a local daemon, $UFO^2$ enables users to issue natural language requests that are translated into coordinated workflows spanning multiple GUI applications. The system provides core abstractions for orchestration, introspection, control execution, and agent collaboration—exposing these as system-level services analogous to those in traditional OSes.

At the heart of $UFO^2$ is a central control plane, the HOSTAGENT, responsible for parsing user intent, managing system state, and dispatching subtasks to a collection of specialized runtime modules called AP-

pAgents. Each AppAgent is dedicated to a particular application (*e.g.*, Excel, Outlook, File Explorer) and encapsulates all logic needed to observe and control that application, including API bindings, UI detectors, and knowledge bases. These modules act as isolated execution contexts with application-specific semantics.

Upon receiving a user request, HostAgent decomposes it into a series of subtasks, each mapped to the application best suited to fulfill it. If the corresponding application is not already running, HostAgent launches it using native Windows APIs and instantiates the corresponding AppAgent. Execution proceeds through a structured loop: each AppAgent continuously observes the application state (via accessibility APIs and vision-based detectors), reasons about the next operation using a ReAct-style planning cycle Yao et al. (2023), and invokes the appropriate action—either a GUI event or a native API call. This loop continues until the subtask terminates, either successfully or due to an unrecoverable error.

UFO$^2$ implements shared memory and control flow via a global blackboard interface, allowing HostAgent and AppAgents to exchange intermediate results, dependency states, and execution metadata. This architecture supports complex workflows across application boundaries—for instance, extracting data from a spreadsheet and using it to populate fields in a web form—without requiring hand-crafted scripts or coordination logic. Crucially, all interactions occur within a virtualized, PiP-based desktop environment, ensuring process-level isolation and safe multi-application concurrency.

**Design Rationale: Centralized Multiagent Runtime.** UFO$^2$ adopts a centralized multiagent Zhang et al. (2024b); Qiao et al. (2023); Han et al. (2024); Wang et al. (2024a) runtime to support both reliability and extensibility. The centralized HostAgent acts as a control plane, simplifying task-level orchestration, error handling, and lifecycle management. Meanwhile, each AppAgent is architected as a loosely coupled executor that encapsulates deep, application-specific functionality.

Modularity at the AppAgent level allows developers and third-party contributors to incrementally expand UFO$^2$'s capabilities by authoring new application interfaces and API bindings. These agents are discoverable, self-contained, and dynamically instantiated by the runtime as needed. From a security and evolvability perspective, this separation of concerns ensures that application logic can evolve independently of the core task orchestration engine.

Together, the HostAgent –AppAgent model allows UFO$^2$ to function as a scalable, pluggable runtime substrate for GUI automation—abstracting away the complexity of heterogeneous interfaces and providing a unified system interface to structured application behavior.

## 3.2 HostAgent: System-Level Orchestration and Execution Control

The HostAgent serves as the centralized control plane of UFO$^2$. It is responsible for interpreting user-specified goals, decomposing them into structured subtasks, instantiating and dispatching AppAgent modules, and coordinating their progress across the system. HostAgent provides system-level services for introspection, planning, application lifecycle management, and multi-agent synchronization.

Figure 3 outlines the architecture of HostAgent. Operating atop the native Windows substrate, HostAgent monitors active applications, issues shell commands to spawn new processes as needed, and manages the creation and teardown of application-specific AppAgent instances. All coordination occurs through a persistent state machine, which governs the transitions across execution phases.

**Responsibilities and Interfaces.** HostAgent exposes the following system services:

- **Task Decomposition.** Given a user's natural language input, HostAgent identifies the underlying task goal and decomposes it into a dependency-ordered subtask graph.

- **Application Lifecycle Management.** For each subtask, HostAgent inspects system process metadata (via UIA APIs) to determine whether the target application is running. If not, it launches the program and registers it with the runtime.

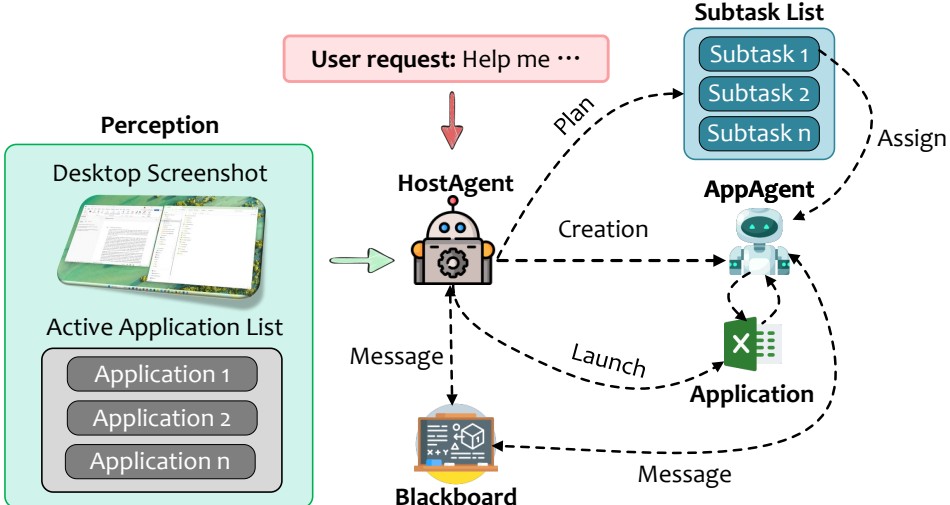

Figure 3: High-level architecture of HOSTAGENT as a control-plane orchestrator.

- **AppAgent Instantiation.** HOSTAGENT spawns the corresponding APPAGENT for each active application, providing it with task context, memory references, and relevant toolchains (*e.g.*, APIs, documentation).

- **Task Scheduling and Control.** The global execution plan is serialized into a finite state machine (FSM), allowing HOSTAGENT to enforce execution order, detect failures, and resolve dependencies across agents.

- **Shared State Communication.** HOSTAGENT reads from and writes to a global blackboard, enabling inter-agent communication and system-level observability for debugging and replay.

**System Perception and Introspection.** To perform its control functions, HOSTAGENT fuses two layers of system introspection:

1. **Visual Layer.** Captures pixel-level screenshots of the desktop workspace, enabling coarse-grained layout understanding.

2. **Semantic Layer.** Queries Windows UIA APIs to extract structural metadata about applications, windows, and control hierarchies.

This dual perception enables HOSTAGENT to resolve ambiguities, detect runtime inconsistencies, and guide agents with context-aware decisions.

**Structured Output Interface.** HOSTAGENT produces structured outputs to drive downstream execution:

- *Subtask Plan:* A high-level execution plan detailing decomposed subtasks.

- *Shell Command:* A sequence of shell-level invocations for managing application lifecycles.

- *Assigned Application:* The process name and index of the application selected for instantiating the APPAGENT, which will be used to execute the next subtask.

- *Agent Messages:* Context-specific instructions passed to APPAGENT instances for localized execution.

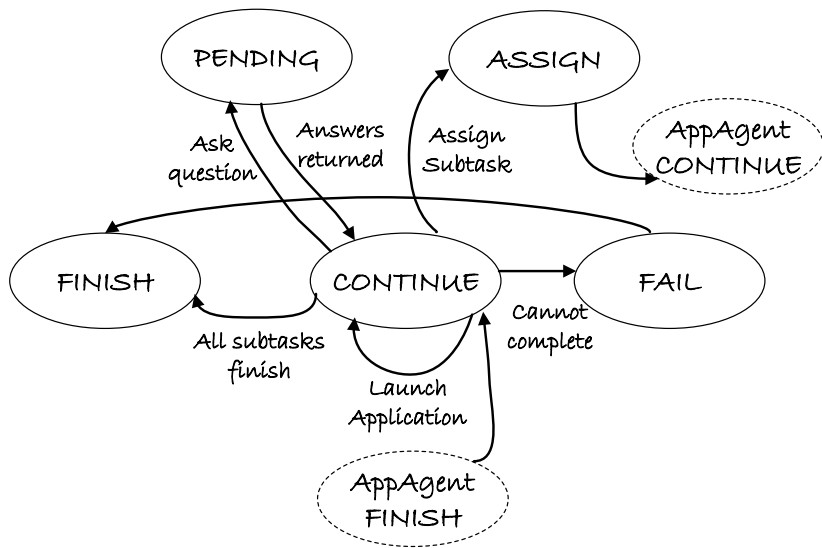

Figure 4: Control-state transitions managed by HOSTAGENT.

- *User Prompts:* Interactive clarification requests in cases of ambiguity or failure.

- *HOSTAGENT State:* The current state within the HOSTAGENT's internal FSM.

**Execution via Finite-State Controller.** The core logic of HOSTAGENT is modeled as a finite-state controller (Figure 4), with the following states:

- `CONTINUE:` Main execution loop; evaluates which subtasks are ready to launch or resume.

- `ASSIGN:` Selects an available application process and spawns the corresponding APPAGENT agent.

- `PENDING:` Waits for user input to resolve ambiguity or gather additional task parameters.

- `FINISH:` All subtasks complete; cleans up agent instances and finalizes session state.

- `FAIL:` Enters recovery or abort mode upon irrecoverable failure.

This explicit FSM structure enables HOSTAGENT to robustly orchestrate dynamic workflows while maintaining high-level guarantees over task completion and fault isolation.

**Memory and State Management.** HOSTAGENT maintains two classes of persistent state:

- **Private State:** Tracks user intent, plan progress, and the control flow of the current session.

- **Shared Blackboard:** A concurrent, append-only memory space that facilitates transparent agent communication by recording key observations, intermediate results, and execution metadata accessible to all APPAGENT instances.

This separation ensures that local context remains encapsulated, while global coordination is visible and consistent across the system.This separation ensures that each agent retains clean, scoped state while benefiting from a globally consistent view for collaborative task execution.

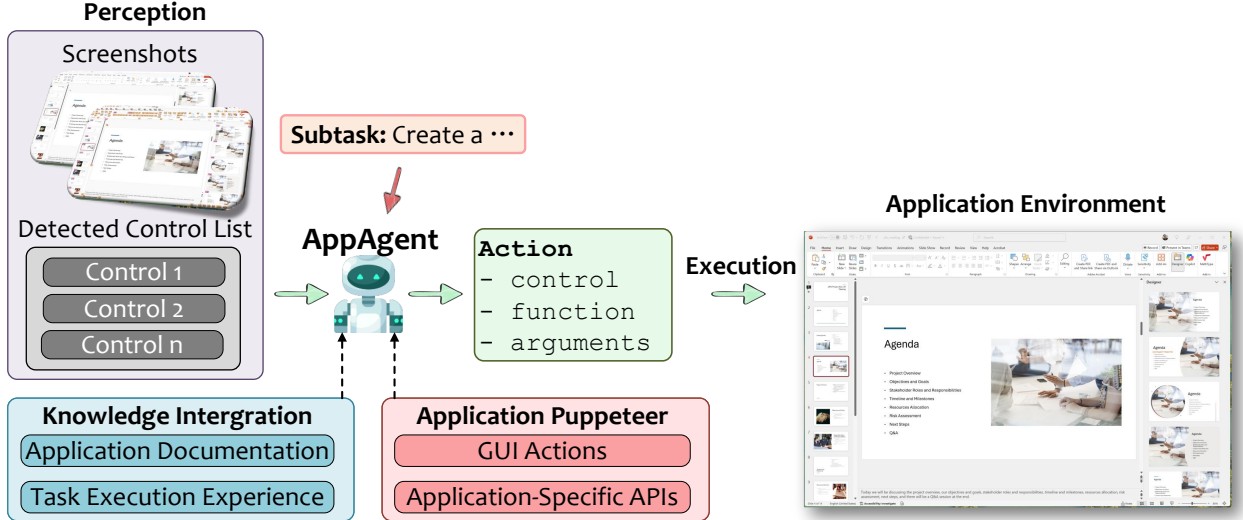

Figure 5: Architecture of an AppAgent, the per-application execution runtime in UFO$^2$.

**Role of HostAgent and Shared Blackboard.** The HostAgent and the shared blackboard serve different but complementary roles in UFO$^2$. The HostAgent acts as the centralized control plane: it decomposes a user request into application-level subtasks, selects or launches the appropriate applications, instantiates AppAgents, schedules subtask execution, and monitors progress through an explicit finite-state controller. Without this control plane, a desktop agent must directly solve the entire user request within a single execution context, which increases ambiguity in application selection, weakens lifecycle management, and makes cross-application workflows difficult to coordinate.

The shared blackboard provides the state-sharing substrate for multi-application execution. It records intermediate artifacts, dependency states, execution metadata, errors, and messages produced by HostAgent and AppAgents. This design is especially important for cross-application tasks, where the output of one AppAgent, such as a table extracted from Excel or a file generated in PowerPoint, becomes the input to another AppAgent. The blackboard therefore enables structured handoff and replayable provenance, rather than relying on implicit conversation history or ad-hoc text messages. Together, HostAgent orchestration and blackboard-based state sharing allow UFO$^2$ to scale from single-application automation to compositional desktop workflows.

Overall, HostAgent abstracts away the complexity of managing concurrent, stateful, cross-application workflows in desktop environments Zhang et al. (2024e). Its control-plane role enables modular execution, coordinated progress, and robust task lifecycle management—all critical features in scaling desktop automation to real-world deployments.

### 3.3 AppAgent: Application-Specialized Execution Runtime

The AppAgent is the core execution runtime in UFO$^2$, responsible for carrying out individual subtasks within a specific Windows application. Each AppAgent functions as an isolated, application-specialized worker process launched and orchestrated by the central HostAgent (Section 3.2). Unlike monolithic CUAs that treat all GUI contexts uniformly, each AppAgent is tailored to a single application and operates with deep knowledge of its API surface, control semantics, and domain logic.

Figure 5 outlines the architecture of an AppAgent. Upon receiving a subtask and execution context from the HostAgent, the AppAgent initializes a ReAct-style control loop Yao et al. (2023), where it iteratively senses the current application state, reasons about the next step, and executes either a GUI or API-based action. This hybrid execution layer—implemented via a `Puppeteer` interface—enables reliable control over dynamic and complex UIs by favoring structured APIs whenever available, while retaining fallback to GUI-based interaction when necessary.

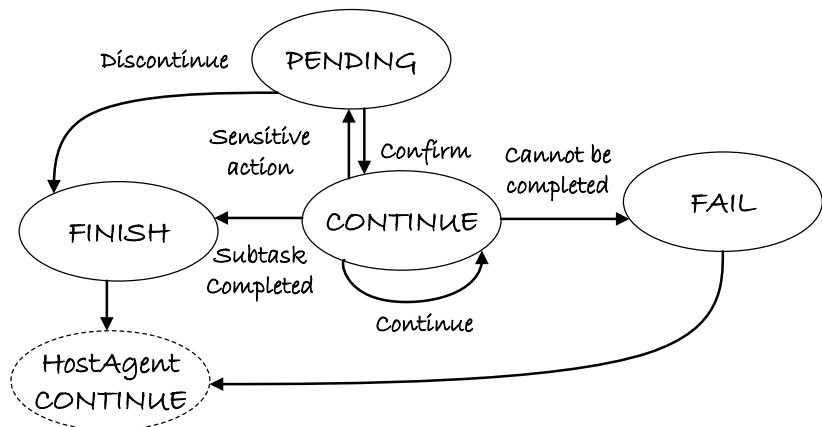

Figure 6: Control-state transitions for an APPAGENT runtime.

**Perception Layer.** Each APPAGENT fuses multiple streams of perception:

- **Visual Input:** Captures GUI screenshots for layout understanding and control grounding.

- **Semantic Metadata:** Extracted from Windows UIA APIs, including control type, label, hierarchy, and enabled state.

- **Symbolic Annotation:** Uses Set-of-Mark (SoM) techniques Yang et al. (2023a) to annotate the control on screenshots.

These fused signals are converted into a structured observation object containing both the GUI screenshot and the set of candidate control elements. This multi-modal representation enables a more comprehensive understanding of the application state, going well beyond raw visual input alone.

**Structured Output.** Based on this state, the APPAGENT produces a structured output:

- Target control (if applicable)

- Action type (*e.g.*, click, type, call API)

- Arguments or payload

- Reasoning trace and planning with Chain-of-Thought (CoT) Wei et al. (2022); Ding et al. (2024)

- Current state in local FSM

This design decouples perception from actuation, enabling deterministic replay, offline debugging, and fine-grained observability.

**Execution via Finite-State Controller.** Each APPAGENT maintains a local finite-state machine (Figure 6) that governs its behavior within the assigned application context:

- `CONTINUE`: Default state for action planning and execution.

- `PENDING`: Invoked for safety-critical actions (*e.g.*, destructive operations); requires user confirmation.

- `FINISH`: Task completed; execution ends.

- `FAIL`: Irrecoverable failure detected (*e.g.*, application crash, permission error).

This bounded execution model isolates failures to the current task and enables safe preemption, retry, or delegation. The FSM also supports interruptible workflows, which can resume from intermediate checkpoints.

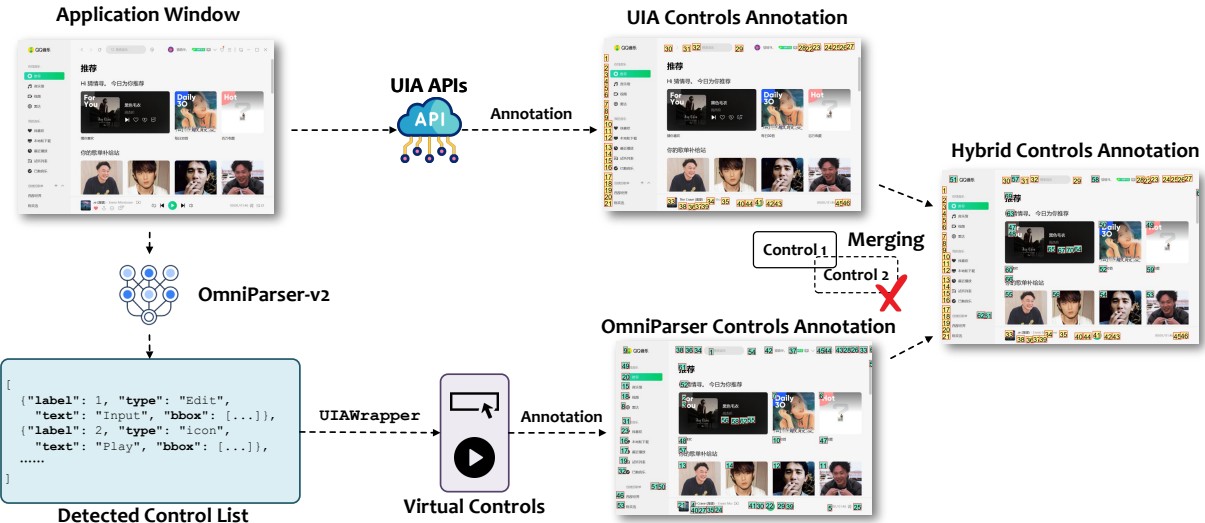

Figure 7: The hybrid control detection approach employed in UFO$^2$.

**Memory and State Coordination.** To enable stateful execution and maintain contextual awareness, each APPAGENT maintains:

- **Private State:** A local log of all executed actions, control decisions, and CoT traces.

- **Shared State:** Updates to the system-wide blackboard, including intermediate outputs, encountered errors, and application-level insights.

This dual-memory design enables APPAGENTs to act autonomously on behalf of HOSTAGENT while remaining synchronized with the broader system. It also supports composability: one APPAGENT's output may become another's input in downstream subtasks.

**Application-Aware SDK and Extensibility.** To support rapid onboarding of new applications, UFO$^2$ exposes an SDK that encapsulates the development and maintenance of APPAGENTs. Developers can register application-specific APIs via a declarative interface, including function metadata, argument schemas, and prompt bindings. Domain-specific help documents and patch notes can be ingested into a searchable knowledge base that agents query at runtime.

This modular abstraction allows third-party vendors or power users to extend UFO$^2$'s capabilities without retraining any models. New functionality can be integrated by updating the application's APPAGENT module, isolating changes from the rest of the system and minimizing regression risk.

**Summary.** As a per-application execution runtime, each APPAGENT provides modular, domain-aware control that surpasses generic GUI agents in both efficiency and robustness. Its hybrid perception-action loop, plugin-based extensibility, and local fault containment enable UFO$^2$ to scale to large application ecosystems with minimal system-wide disruption.

## 3.4 Hybrid Control Detection

Reliable perception of GUI elements is fundamental to enabling APPAGENTs to interact with application interfaces in a deterministic and safe manner. However, real-world GUI environments exhibit substantial heterogeneity: some applications expose well-structured accessibility data via Windows UI Automation (UIA) APIs, while others—especially legacy or custom applications—render critical controls using non-standard toolkits that bypass UIA entirely.

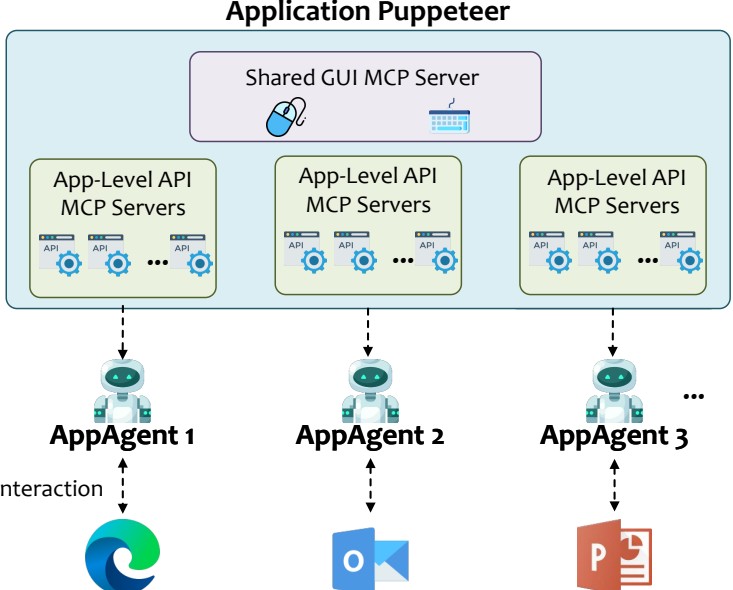

Figure 8: `Puppeteer` serves as a unified execution engine that harmonizes GUI actions and native API calls with MCP servers.

To address this disparity, UFO$^2$ introduces a *hybrid control detection* subsystem that fuses UIA-based metadata with vision-based grounding Lu et al. (2024); Cheng et al. (2024); Gou et al. (2024) to construct a unified and comprehensive control graph for each application window (Figure 7). This design ensures both coverage and reliability, forming a resilient perceptual foundation for downstream action planning and execution.

**UIA-Layer Detection.** When available, UIA offers a semantically rich and high-precision interface for enumerating on-screen controls. The detection pipeline first queries the accessibility tree to extract controls satisfying a set of runtime predicates (*e.g.*, `is_visible()`, `is_enabled()`). These controls are annotated with their attributes (type, label, bounding box) and assigned stable identifiers, forming the initial control graph.

**Vision-Layer Augmentation.** To augment the perception pipeline for UI-invisible or custom-rendered controls, we integrate OmniParser-v2 Lu et al. (2024), a vision-based grounding model designed for fast and accurate GUI parsing. OmniParser-v2 combines a lightweight YOLO-v8 Reis et al. (2023) detector with a fine-tuned Florence-2 (0.23B) Xiao et al. (2024) encoder to process raw application screenshots and identify additional interactive elements. Each detection includes the control type, confidence score, and spatial bounding box.

**Fusion and Deduplication.** We unify these two streams by performing deduplication based on bounding-box overlap. Visual detections with Intersection-over-Union (IoU) greater than 10% against any UIA-derived control are discarded. Remaining visual-only detections are converted into pseudo-UIA objects using a lightweight `UIAWrapper` abstraction, allowing them to seamlessly integrate into the rest of the APPAGENT pipeline. This fused control set is passed downstream to the SoM-based annotation module Yang et al. (2023a). Figure 7 shows a typical scenario involving a hybrid-rendered GUI. Yellow bounding boxes denote standard UIA-detected elements, while blue bounding boxes represent visual-only detections. Both are integrated into a single actionable control graph consumed by the APPAGENT.

```
1  @mcp.tool(tags={"ExcelAgent"})
2  def summarize_table(
3      sheet_name: Annotated[str, Field(description="Name or index of the target
          worksheet")],
4      table_range: Annotated[str, Field(description="Cell range in Excel A1 notation
          (e.g., 'A1:D10')")]
5  ) -> Annotated[Dict, Field(description="Summary statistics of the selected table")
      ]:
6      """
7      Extracts the content of a table from Excel and returns basic
8      statistics (row count, column count, column headers) along with
9      a Markdown preview of the data.
10     """
11     ...
```

Figure 9: An example API registration for Excel. Developers expose high-level operations (here, table summarization) by attaching an `@mcp.tool` decorator. The tool is automatically integrated into the APPA-GENT's runtime action space for Excel.

## 3.5 Unified GUI–API Action Orchestrator

APPAGENTs interact with application environments that typically expose two complementary classes of interfaces: *(i) GUI frontends*, which are universally available but inherently brittle; and *(oi) native APIs*, which offer high-fidelity control but require explicit integration and maintenance Zhang et al. (2025).

To unify these heterogeneous execution backends under a single abstraction, $UFO^2$ introduces the `Puppeteer`, a modular action orchestrator that manages a collection of *Model Context Protocol* (MCP) servers exposing both GUI and API operations. We choose MCP servers for GUI and API integration due to their ease of use, ability to establish a standardized action interface, and strong ecosystem support.

Concretely, `Puppeteer` maintains a shared *GUI MCP server* that simulates generic user inputs such as mouse clicks and keyboard events. This server is accessible to all applications, ensuring coverage for tasks where no native APIs are available. In addition, each application can define its own *API MCP servers* Hou et al. (2025), exposing application-specific capabilities (*e.g.*, Excel cell operations, Outlook email handling). Multiple servers can be registered for a single application, allowing fine-grained decomposition of functionality.

At runtime, `Puppeteer` dynamically selects between GUI-level automation and API-level execution for each action step (Figure 8). This design improves *robustness* (by falling back to GUI actions when APIs are missing), *latency* (by collapsing long GUI sequences into single API calls), and *maintainability* (by modularizing app-specific APIs behind a stable interface). For example, formatting a block of cells in Excel can be executed as one API call instead of a series of GUI interactions, substantially reducing both execution time and the surface area for error Song et al. (2024); Zhang et al. (2025).

To support extensibility, `Puppeteer` provides a lightweight API registration mechanism. Developers can expose high-level operations by attaching simple Python decorators to functions (Figure 9). Each registered function is automatically wrapped with metadata, including its name, argument schema, and associated APPAGENT binding and incorporated into the APPAGENT's runtime action space. This declarative mechanism lowers the barrier for integrating new capabilities, enabling continuous evolution of application-specific automation while preserving architectural modularity and reuse.

At runtime, $UFO^2$ prompts APPAGENT to employs a decision-making policy to choose the most appropriate execution path for each operation. If a semantically equivalent API is available, `Puppeteer` prefers it over GUI automation for reliability and atomicity. If an API fails or is unavailable (*e.g.*, missing bindings or runtime permission errors), the system gracefully falls back to GUI-based control via simulated clicks or keystrokes. This runtime flexibility allows APPAGENT to preserve robustness across heterogeneous environments without sacrificing generality.

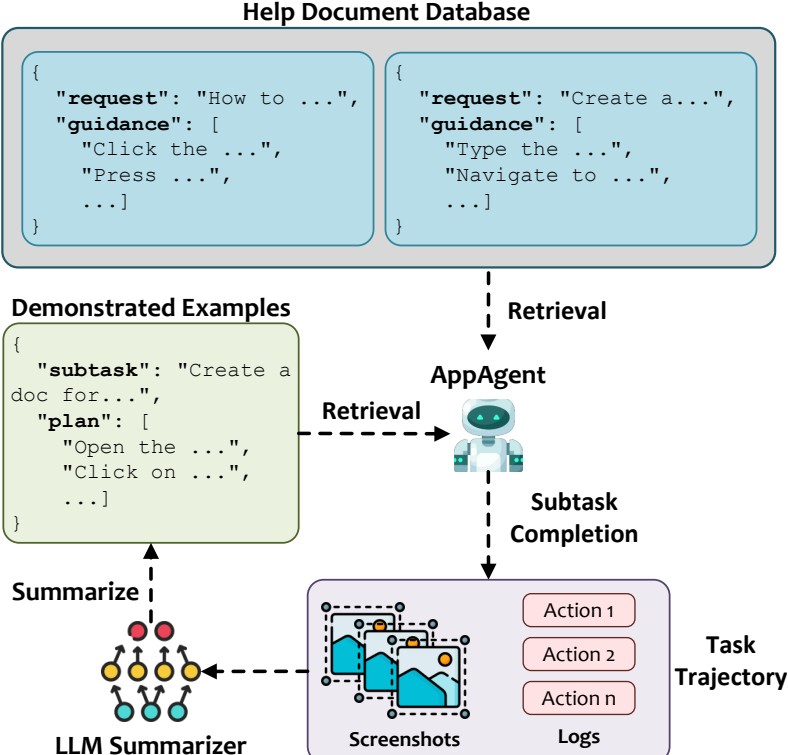

Figure 10: Overview of the knowledge substrate in UFO$^2$, combining static documentation with dynamic execution history.

**Puppeteer** transforms action execution in UFO$^2$ from a monolithic GUI-centric model into a flexible, OS-integrated control layer that mixes perceptual agility with semantic precision. This hybrid execution model not only improves system performance and stability but also lays the groundwork for sustainable integration of application-specific capabilities in future desktop agents.

### 3.6 Continuous Knowledge Integration Substrate

Unlike traditional CUAs, which rely heavily on static training corpora, UFO$^2$ introduces a persistent and extensible *knowledge substrate* that supports runtime augmentation of application-specific understanding. As illustrated in Figure 10, this substrate enables each APPAGENT to retrieve, interpret, and apply external documentation and prior execution traces without requiring retraining of the underlying models. This hybrid memory design functions analogously to an OS-level metadata manager, abstracting over two key knowledge flows: static references (*e.g.*, user manuals) and dynamic experience (*e.g.*, execution logs).

**Bootstrapping from Documentation.** Most real-world desktop applications expose substantial task-level documentation via user guides, help menus, or online tutorials. UFO$^2$ capitalizes on this resource by offering a one-click interface to parse and ingest such documents into an application-specific vector store. Documents are structured as `json` records with a natural language description in the `request` field and detailed execution guidance in the `guidance` field.

At runtime, when an APPAGENT receives a subtask, it queries this indexed store to retrieve relevant guidance, which is then injected into the agent's prompt. This mechanism effectively mitigates cold-start issues—especially when handling novel applications or infrequent operations—by enriching the agent's reasoning context with domain-grounded procedural knowledge.

---

**Algorithm 1** Speculative Multi-Action Execution in $\text{UFO}^2$

---

**Require:** Initial UI context $C_0$, batch size $k$
**Ensure:** List `Executed` of actions completed so far
    **Stage 1: Batch Prediction**
1: $A \leftarrow \text{LLM\_PREDICT}(C_0, k)$                                        $\triangleright\ A = \left[(ctrl_i, op_i)\right]_{i=1}^{k}$
2: `Executed` $\leftarrow$ [ ];     $C \leftarrow C_0$
    **Stage 2 & 3: Sequential Validate-Execute Loop**
3: **for** $i \leftarrow 1$ **to** $k$ **do**
4:     $(ctrl, op, \_) \leftarrow A[i]$
      *// Validate in the* current *context*
5:     **if not** $\text{UIA\_ISENABLED}(ctrl, C) \lor$ **not** $\text{UIA\_ISVISIBLE}(ctrl, C)$ **then**
6:         **break**                                   $\triangleright$ validation failed $\rightarrow$ early stop
7:     **end if**
      *//Execute and refresh context*
8:     $\text{EXECUTE}(ctrl, op)$
9:     append $(ctrl, op)$ to `Executed`
10:    $C \leftarrow \text{UIA\_GETCONTEXT}()$                                   $\triangleright$ UI changed
11: **end for**
12: **if** $|\text{Executed}| < k$ **then**
13:    $\text{REPORTPARTIAL}(\text{Executed})$
14:    $\text{REPLAN}(C)$
15: **end if**

---

**Reinforcing from Experience.** Beyond static knowledge, $\text{UFO}^2$ continuously learns from its own execution history. Each automation run produces structured logs—including natural language task descriptions, executed action sequences, application screenshots, and final outcomes. Periodically, these logs are mined offline by a summarization module that distills successful trajectories into reusable `Example` records.

Each record contains a task signature and an associated step-by-step plan, stored in an application-specific example database. When a similar task is encountered in the future, APPAGENT uses In-Context Learning (ICL) Dong et al. (2024); Min et al. (2022); Zhang et al. (2024c); Luo et al. to retrieve relevant demonstrations and improve execution fidelity. This dynamic reinforcement pipeline transforms the system into a long-lived agent that improves with use, without introducing the brittleness or operational cost of fine-tuning Wang et al. (2025).

**Runtime RAG Integration.** At the system level, the knowledge substrate acts as a Retrieval-Augmented Generation (RAG) layer Lewis et al. (2020); Liu et al. (2024); Gao et al. (2023); Jiang et al. (2024) that bridges the gap between pre-trained language models and application-specific requirements. Because both help documents and examples are indexed with semantic embeddings, the retrieval pipeline is fast, interpretable, and cache-friendly. Additionally, versioned indexing ensures that knowledge artifacts can evolve alongside software updates, preventing model obsolescence and supporting robust execution across long deployment cycles.

By integrating static and experiential knowledge into a unified RAG pipeline, $\text{UFO}^2$ transforms CUAs from brittle, training-time constructs into dynamic, evolving agents. This substrate plays a foundational role in enabling sustainable automation across complex, heterogeneous application ecosystems.

### 3.7 Speculative Multi-Action Execution

Conventional CUAs suffer from a fundamental execution bottleneck: each automation step is executed in isolation, requiring a full LLM inference for every single GUI action. This *step-wise inference loop* introduces excessive latency, inflates system resource usage, and increases cumulative error rates—especially when interacting with complex or multi-phase workflows Zhang et al. (2025). The root cause is the dynamic and

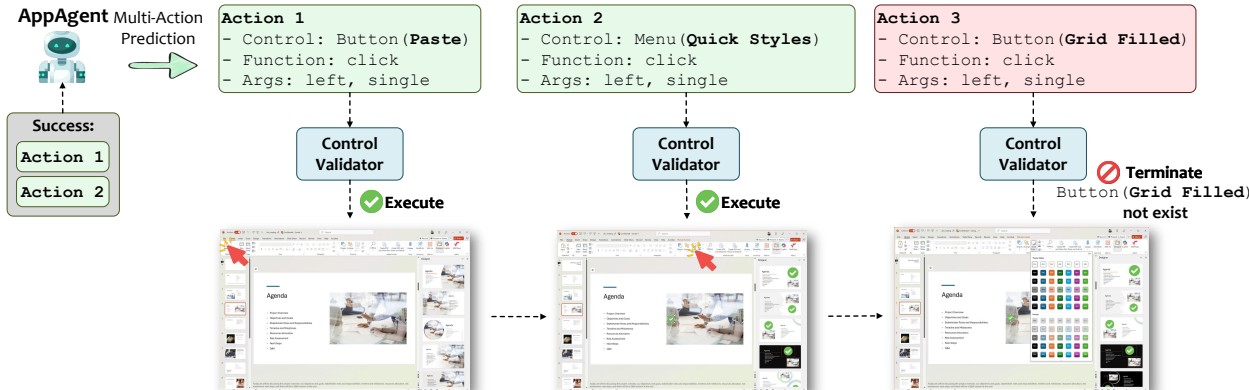

Figure 11: Speculative multi-action execution in UFO$^2$: batched inference with online validation.

uncertain nature of GUI environments, where any single action may alter the interface and invalidate future plans.

To overcome these limitations, UFO$^2$ introduces a system-level optimization called *speculative multi-action execution*, inspired by classical ideas from speculative execution in processor design and instruction pipelining. Rather than issuing one action per LLM call, UFO$^2$ speculatively generates a *batch* of likely next steps using a single inference pass and validates their applicability at runtime through tight OS integration. We present an algorithm in 1.

The speculative executor operates in three stages:

1. **Action Prediction:** The APPAGENT issues a single LLM query to predict multiple plausible actions under its current context. Each predicted step includes a target control, intended operation, and rationale.

2. **Runtime Validation:** For each action, the system consults the Windows UIA API to verify the action's preconditions (*e.g.*, `is_enabled()`, `is_visible()`). This check ensures that each target control is still valid and interactive.

3. **Sequential Execution and Early Exit:** Actions are executed in order, halting immediately if any validation fails due to interface change (*e.g.*, control no longer exists or is disabled). The executor then reports a partial result set and prompts the agent to replan.

We show an illustrative example of speculative multi-action execution in Figure 11. In this case, the APPAGENT initially plans to execute three actions in a single step: clicking `Paste`, then `Quick Style`, and finally `Grid Filled`. However, after the second action, the control validator detects that the control required for the third action (`Grid Filled`) is no longer present—likely because the GUI layout changed as a result of the previous step. The `Puppeteer` then terminates execution at that point and returns the partial results. This example highlights how UFO$^2$ safely handles speculative execution by validating each control before acting, ensuring robustness even in the face of dynamic interface changes.

Overall, this strategy drastically reduces LLM invocation frequency and amortizes the cost of action planning across multiple steps, while preserving the correctness guarantees of per-step validation. Critically, validation is performed by trusted OS-level APIs instead of vision models, ensuring high reliability and eliminating spurious interactions.

## 4 Picture-in-Picture Interface

A key design objective of UFO$^2$ is to deliver high-throughput automation while preserving the responsiveness and usability of the primary desktop environment. Existing CUAs often monopolize the user's workspace,

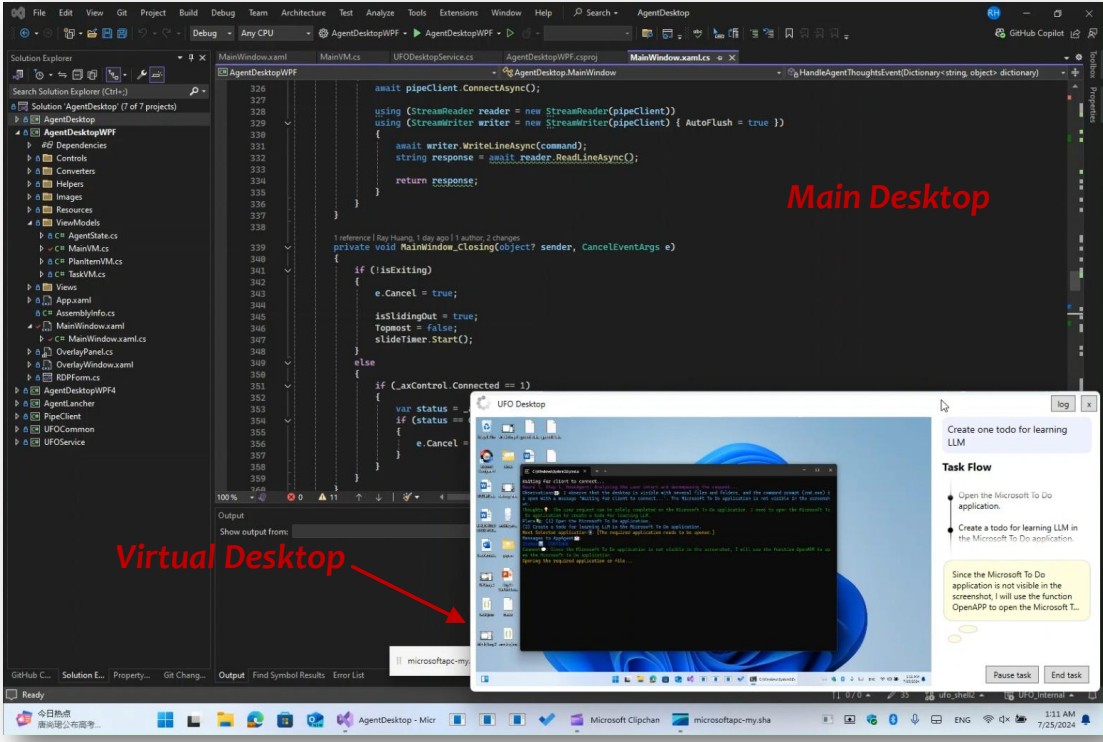

Figure 12: The Picture-in-Picture interface: a virtual desktop window for non-disruptive automation.

seizing mouse and keyboard control for extended periods and making the system effectively unusable during task execution. To overcome this, UFO$^2$ introduces a *Picture-in-Picture* (PiP) interface: a lightweight, virtualized desktop window powered by Remote Desktop loopback, enabling fully isolated agent execution in parallel with active user workflows, as illustrated in Figure 12.

## 4.1 Virtualized User Environment with Minimal Disruption

Unlike conventional CUAs that operate in the main desktop session, the PiP interface presents a resizable, movable window containing a fully functional replica of the user's desktop. Internally, this is implemented via Windows' native Remote Desktop Protocol (RDP) loopback Miller & Pegah (2007), creating a distinct virtual session hosted on the same machine. Applications launched within the PiP session inherit the user's identity, credentials, settings, and network context, ensuring consistency with foreground operations.

From the user's perspective, the PiP window behaves like a sandboxed workspace: the automation executes in the background, visible but unobtrusive. The user retains full control of the primary desktop and can minimize or reposition the PiP window at will. This enables UFO$^2$ to perform long-running or repetitive workflows (*e.g.*, data entry, batch file processing) without blocking user interaction or degrading responsiveness.

## 4.2 Robust Input and State Isolation

To ensure robust separation between agent actions and user activities, UFO$^2$ leverages the RDP subsystem to maintain distinct input queues and device contexts across sessions. Mouse and keyboard events generated within the PiP desktop are fully scoped to that session and cannot interfere with the primary desktop. Similarly, GUI changes and focus transitions are restricted to the virtual environment.

This level of input isolation is critical for preventing accidental interference—either by the user or the agent—and ensures that automation sequences remain stable, even during simultaneous foreground activity. The architecture also supports controlled error recovery: failures or unexpected UI states within the PiP session do not propagate to the primary desktop, preserving the integrity of the user's environment.

### 4.3 Secure Cross-Session Coordination

Although visually and operationally distinct, the PiP session must remain logically connected to the host environment. To enable this, UFO$^2$ establishes a secure inter-process communication (IPC) channel between the PiP agent runtime and a host-side coordinator. We implement this using Windows Named Pipes, authenticated and encrypted using per-session credentials Venkataraman & Jagadeesha (2015).

This IPC layer supports two-way messaging:

- From the host to the PiP: task assignment, progress polling, cancellation, and user clarifications.

- From the PiP to the host: status updates, completion reports, and exception notifications.

Users interact with the automation pipeline through a lightweight frontend panel on the host desktop, enabling real-time visibility and partial control without needing to directly access the PiP window. This transparent yet secure communication channel ensures trust and usability, particularly in long-running or partially supervised workflows.

### 4.4 System-Level Implications

The PiP interface represents more than a UX refinement—it is a system-level abstraction that reconciles concurrency, usability, and safety. It decouples automation execution from foreground interactivity, introduces a new isolation primitive for GUI-based agents, and simplifies failure recovery by sandboxing side effects. By exploiting existing RDP capabilities with minimal system overhead, the PiP interface offers a practical and backwards-compatible approach to scalable desktop automation.

However, PiP should be viewed as an execution-isolation mechanism rather than a full security sandbox. Applications inside the PiP session may still inherit the user's credentials, file-system access, and network permissions, depending on deployment configuration. Therefore, PiP should be combined with permission control, tool-level restrictions, and audit logging when UFO$^2$ is deployed in sensitive enterprise environments.

## 5 Implementation and Specialized Engineering Design

We implement UFO$^2$ as a full-stack desktop automation framework spanning over 30,000 lines of `Python` and `C#` code. Python serves as the core runtime environment for agent orchestration, control logic, and API integration, while `C#` supports GUI development, debugging interfaces, and Windows-specific operations such as the Picture-in-Picture desktop. To support retrieval-augmented reasoning, UFO$^2$ leverages Sentence Transformers Reimers & Gurevych (2019) for embedding-based document and experience retrieval.

Beyond its core functionality, UFO$^2$ incorporates multiple specialized engineering components that target critical systems goals: composability, interactivity, debuggability, and scalable deployment. We highlight several key mechanisms below.

### 5.1 Multi-Round Task Execution

Unlike stateless one-shot agents, UFO$^2$ adopts a session-based execution model to support iterative, interactive workflows (Figure 13). Each `Session` maintains persistent contextual memory—including intermediate results, task progress, and application state—across multiple `Rounds` of execution. Users can refine prior instructions, launch follow-up tasks, or intervene when agents encounter ambiguous or unsafe operations.

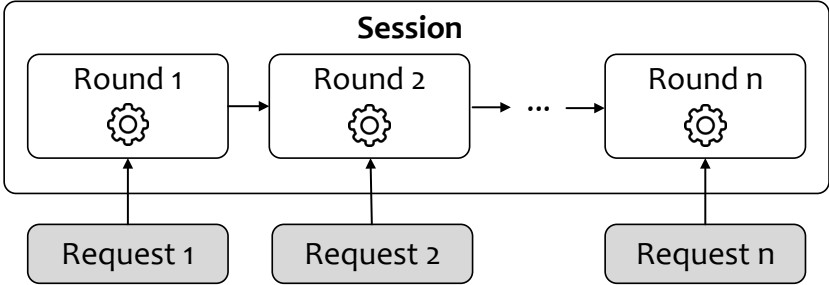

Figure 13: The interactive `Session` model in UFO$^2$ supports multi-round refinement.

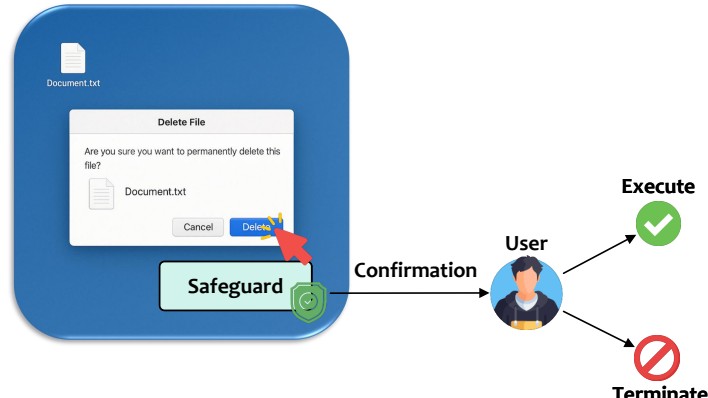

Figure 14: The safeguard mechanism employed in UFO$^2$.

This multi-round interaction paradigm facilitates progressive convergence on complex tasks while preserving transparency and human oversight. It enables UFO$^2$ to support human-in-the-loop refinement strategies, bridging static LLM workflows with dynamic user guidance.

## 5.2 Safeguard Mechanism

While automation substantially boosts productivity, any CUA carries inherent risks of executing unsafe actions that may adversely affect user data or system stability Levy et al. (2024); Zhang et al. (2024b). Examples include deleting critical files, terminating applications prematurely (resulting in unsaved data loss), or activating sensitive devices such as webcams without explicit consent. These actions pose severe risks, potentially causing irrecoverable damage or security breaches.

To mitigate such risks, UFO$^2$ incorporates an explicit *safeguard mechanism*, designed to actively detect potentially dangerous actions, as shown in Figure 14. Specifically, whenever an APPAGENT identifies an action matching predefined risk criteria, it transitions into a dedicated `PENDING` state, pausing execution and actively prompting the user for confirmation. Only upon receiving explicit user consent does the agent proceed; otherwise, the action is aborted to prevent harm. The definition and scope of what constitutes a risky action are fully customizable through a straightforward prompt-based interface, enabling users and system administrators to precisely tailor safeguard behavior according to their organization's specific risk policies. This flexibility allows the safeguard system to be dynamically adapted as automation requirements evolve.

Through this proactive safety-checking framework, UFO$^2$ significantly reduces the likelihood of executing harmful operations, thus enhancing overall system safety, user trust, and robustness in real-world deployments.

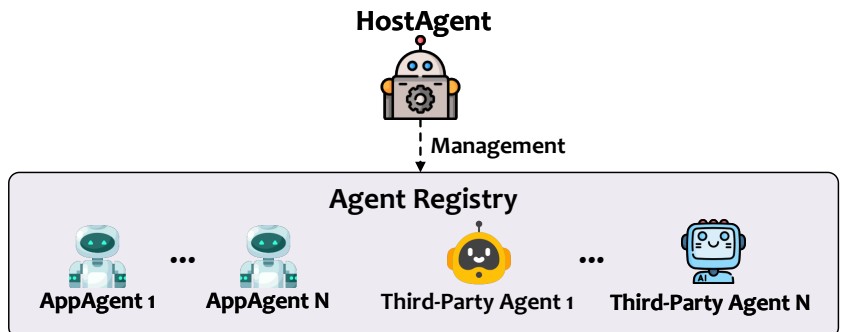

Figure 15: The agent registry supports seamless wrapping of third-party components into the APPAGENT framework.

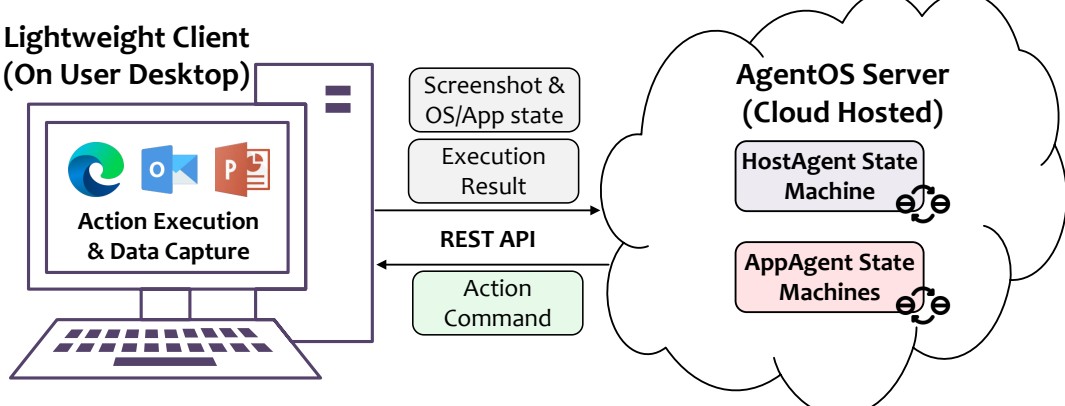

Figure 16: The client-server deployment model used in AgentOS-as-a-Service.

### 5.3 Everything-as-an-AppAgent

To support ecosystem extensibility, UFO$^2$ introduces an agent registry mechanism that encapsulates arbitrary third-party components as pluggable APPAGENTs (Figure 15). Through a simple registration API, external automation solutions—such as domain-specific copilots or proprietary tools—can be wrapped with lightweight compatibility shims that expose a unified interface to the HOSTAGENT.

This design enables HOSTAGENT to treat native and external APPAGENTs interchangeably, dispatching subtasks based on capability and specialization. We find that even minimal wrappers (*e.g.*, for OpenAI Operator OpenAI (2025)) lead to tangible performance gains, highlighting the system's modularity and its ability to incorporate diverse execution backends with minimal engineering overhead.

### 5.4 AgentOS-as-a-Service

UFO$^2$ adopts a client-server architecture to support practical deployment at scale (Figure 16). A lightweight client resides on the user's machine and is responsible for GUI operations and application-side sensing. Meanwhile, a centralized server (running on-premises or in the cloud) hosts the HOSTAGENT/APPAGENT logic, orchestrates workflows, and handles LLM queries.

This separation of control and execution offers several systems-level benefits:

- **Security**: Sensitive orchestration and model execution are isolated from user devices.

- **Maintainability**: Server-side updates propagate without modifying the client.

**Markdown-Formatted Log Viewer**                    **Debugging Tool**

Figure 17: An illustration of the markdown-formatted log viewer and debugging tool in UFO$^2$.

- **Scalability**: The system can support multiple concurrent clients with centralized scheduling and load management.

The client-server boundary enforces a clean service abstraction, promoting modularity and simplifying rollout in enterprise environments, and separates environment interaction from agent reasoning. The lightweight desktop client captures screenshots and OS/application state, executes approved actions, and reports results, while the server hosts HostAgent/AppAgent logic and model calls. This separation simplifies centralized updates and enterprise deployment, but it does not by itself remove all scalability bottlenecks. In practice, concurrent throughput is constrained by LLM inference latency, model API rate limits, network round-trip time, and the server's scheduling capacity across active sessions.

Accordingly, we treat the client-server architecture as a deployment abstraction rather than a claim of unlimited scalability. Future large-scale deployments should profile per-client CPU and memory overhead, server-side scheduling latency, model-call throughput, and failure recovery under network interruptions.

## 5.5 Comprehensive Logging and Debugging Infrastructure

Robust observability is essential for diagnosing failures and supporting ongoing system improvement. To this end, UFO$^2$ implements a comprehensive logging and debugging framework. Each session captures fine-grained traces of execution: prompts, LLM outputs, control metadata, UI state snapshots, and error events.

At the end of each session, UFO$^2$ compiles these artifacts into a structured, Markdown-formatted execution log. Developers can inspect action-by-action agent decisions, visualize interface state transitions, and replay behavior for debugging. The framework also supports prompt editing and selective replay for targeted hypothesis testing, significantly accelerating the debugging cycle. We show an example of these tools in Figure 17.

This observability layer functions as a lightweight provenance system for agent behavior, fostering transparency, accountability, and rapid iteration during deployment.

## 5.6 Automated Task Evaluator

To provide structured feedback and facilitate continuous improvement, UFO$^2$ includes an automated task evaluation engine based on LLM-as-a-judge Chen et al. (2024a). As shown in Figure 18, the evaluator parses session traces—including actions, rationales, and screenshots—and applies CoT reasoning to decompose tasks into evaluation criteria.

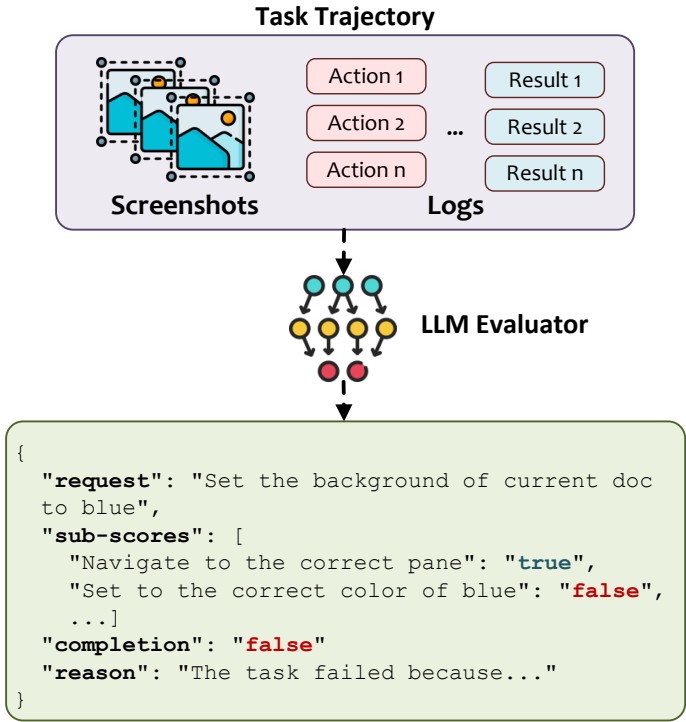

Figure 18: The LLM-based task evaluator applies CoT reasoning to structured session logs.

It assigns partial scores and synthesizes an overall result: *success*, *partial*, or *failure*. This structured outcome feeds into downstream dashboards and debugging tools. It also supports self-monitoring and offline analysis of failure cases, closing the loop between execution, diagnosis, and improvement.

**Summary.** These engineering components demonstrate UFO$^2$'s commitment to operational robustness and extensibility. From session-based execution and pluggable agents to service-oriented deployment and observability infrastructure, each module reflects a design focused on bridging conceptual LLM agent architectures with the systems realities of deployment at scale.

## 6 Evaluation

We tested UFO$^2$ rigorously across more than 20 Windows applications, including office suites, file explorers, and custom enterprise tools to assess performance, efficiency, and robustness. Our experiments show:

1. UFO$^2$ achieves a *10%* higher task completion rate, a 50% relative improvement—over the best-performing current CUA Operator, enabled by deeper OS-level integration.

2. The hybrid UIA–vision approach identifies custom or nonstandard GUI elements missed by UIA alone, boosting success in interfaces with proprietary widgets.

3. Allowing APPAGENTs to invoke native APIs or GUI interactions to improve completion rate by over 8%, cuts latency and reduces the fragility seen in purely click-based workflows.

4. Leveraging external documents and execution logs increases UFO$^2$'s ability to handle unfamiliar features without retraining.

5. Speculative multi-action execution consolidates multiple steps into a single LLM call, lowering inference cost by up to *51.5%* without compromising reliability.

6. By enabling Everything-as-an-AppAgent (*e.g.*, Operator), UFO$^2$ both boosts overall performance and uncovers the full potential of each individual agent.

Overall, these results confirm that UFO$^2$'s deeper integration with Windows and application-level APIs yields both higher performance and reduced overhead, making a compelling case for an OS-native approach to desktop automation.

## 6.1 Experimental Setup

**Deployment Environment.** The benchmark environments are hosted on isolated VMs with *8* AMD Ryzen 7 CPU cores and *8* GB of memory, matching typical deployment conditions. All GPT-family models (GPT-4o, o1, and Operator) are accessed via Azure OpenAI services, while the OmniParser-v2 and Qwen-3.5 Team (2026) vision model operates on a separate virtual machine provisioned with an NVIDIA A100 80GB GPU to support efficient and high-throughput visual grounding.

**Benchmarks.** We evaluate UFO$^2$ using two established Windows-centric automation benchmarks:

- **Windows Agent Arena (WAA) Bonatti et al.:** Consists of 154 live automation tasks across 15 commonly used Windows applications, including office productivity tools, web browsers, system utilities, development environments, and multimedia apps. Each task includes a custom verification script for automated correctness checking.

- **OSWorld-W Xie et al. (2024):** A targeted subset of the OSWorld benchmark specifically tailored for Windows, comprising 49 live tasks across office applications, browser interactions, and file-system operations. Tasks are similarly equipped with handcrafted verification scripts for reliable outcome validation.

Each task runs independently, and verification strictly follows the original scripts provided by each benchmark.[1]

**Baselines.** We compare UFO$^2$ with five representative state-of-the-art CUAs, each leveraging GPT-4o as the inference engine:

- **UFO** Zhang et al. (2024b): A pioneering multiagent, GUI-focused automation system designed explicitly for Windows, integrating UIA and visual perception.

- **NAVI** Bonatti et al.: A single-agent baseline from WAA, utilizing screenshots and accessibility data for GUI understanding.

- **OmniAgent** Lu et al. (2024): Employs OmniParser for visual grounding combined with GPT-based action planning.

- **Agent S** Agashe et al.: Features a multiagent architecture with experience-driven hierarchical planning, optimized for complex, multi-step tasks.

- **Operator** OpenAI (2025): A recent, high-performance CUA from OpenAI, simulating human-like mouse and keyboard interactions via screenshots.

These baselines were selected for their representativeness of diverse architectural and design paradigms (*e.g.*, single-agent vs. multiagent, GUI-only vs. hybrid approaches). To ensure fairness, each agent is restricted to a maximum of *30 execution steps* per task, reflecting practical user expectations and preventing excessively long task executions. Additionally, we evaluate a base version of UFO$^2$ (termed UFO$^2$-base) using only

---

[1]Reported baseline scores in OSWorld differ slightly from prior results focused on Ubuntu due to corrections in verification scripts and alignment with Windows-specific tasks (OSWorld-W).

Table 1: Comparison of success rates (SR) across agents on WAA and OSWorld-W benchmarks.

| Agent | Model | WAA | OSWorld-W |
|---|---|---|---|
| UFO | GPT-4o | 19.5% | 12.2% |
| NAVI | GPT-4o | 13.3% | 10.2% |
| OmniAgent | GPT-4o | 19.5% | 8.2% |
| Agent S | GPT-4o | 18.2% | 12.2% |
| Operator | computer-use | 20.8% | 14.3% |
| UFO$^2$-base | GPT-4o | 23.4% | 22.4% |
| UFO$^2$-base | o1 | 25.3% | 22.4% |
| **UFO$^2$** | **GPT-4o** | **27.9%** | **28.6%** |
| **UFO$^2$** | **o1** | **30.5%** | **32.7%** |

UIA detection, GUI-based interactions, and without dynamic knowledge integration, alongside the full implementation of UFO$^2$ featuring hybrid control detection, combined GUI-API interactions, and continuous knowledge augmentation. API integrations were selectively implemented for three office applications within OSWorld-W as illustrative examples; no APIs were introduced for the WAA tasks. Further implementation details are available in Section 6.4.

**Evaluation Metrics.** We utilize two primary metrics for performance evaluation:

- **Success Rate (SR):** Defined as the percentage of tasks successfully completed, validated via the benchmarks' own verification scripts.

- **Average Completion Steps (ACS):** Measures the average number of LLM-involved action inference steps required per task. Fewer steps correspond to higher efficiency, directly correlating with lower inference latency and reduced computational overhead.

These metrics effectively reflect both functional effectiveness and practical efficiency, providing clear indicators of real-world automation performance.

## 6.2 Success Rate Comparison

Table 1 summarizes the success rates (SR) of all evaluated agents across the WAA and OSWorld-W benchmarks, as verified by each benchmark's automated validation scripts. Notably, even the basic configuration (UFO$^2$-base)—which relies solely on standard UI Automation and GUI-driven actions—consistently surpasses prior state-of-the-art CUAs. Specifically, with GPT-4o, UFO$^2$-base achieves an SR of 23.4% on WAA, outperforming the best existing baseline, Operator (20.8%), by 2.6%. This margin widens significantly when employing the stronger o1 LLM, lifting UFO$^2$-base's performance to 25.3%.

Moreover, the complete version of UFO$^2$, incorporating hybrid GUI–API action execution, advanced visual grounding, and continuous knowledge integration, further amplifies these performance gains. With GPT-4o, UFO$^2$ achieves a 27.9% SR on WAA, exceeding Operator by a substantial 7.1%. The performance gap becomes even more pronounced on OSWorld-W, where UFO$^2$ achieves a 28.6% SR compared to Operator's 14.3%, effectively doubling its success rate. Utilizing the stronger o1 model further improves UFO$^2$'s performance to 30.5% (WAA) and 32.7% (OSWorld-W), solidifying its leading position.

These significant performance improvements clearly underscore the advantages of UFO$^2$'s deep integration with OS-level mechanisms and its unified system architecture. While prior CUAs primarily emphasize model-level optimization or singular reliance on visual interfaces, our results demonstrate that robust, system-level orchestration—combining structured OS APIs, specialized application knowledge, and hybrid GUI–API interaction—is instrumental in achieving higher task reliability and broader automation coverage. Crucially, even a general-purpose, less-specialized model like GPT-4o can surpass highly specialized CUAs (such as Operator) when integrated within the comprehensive UFO$^2$ framework. This insight reinforces the value of architectural design and OS integration as key drivers of practical, deployable desktop automation solutions.

Table 2: SR breakdown by application type on WAA and OSWorld-W.

| Agent | Model | WAA | | | | | | OSWorld-W | |
|---|---|---|---|---|---|---|---|---|---|
| | | Office | Web Browser | Windows System | Coding | Media & Video | Windows Utils | Office | Cross-App |
| UFO | GPT-4o | 0.0% | 23.3% | 33.3% | 29.2% | 33.3% | 8.3% | 18.5% | 4.5% |
| NAVI | GPT-4o | 0.0% | 20.0% | 29.2% | 9.1% | 25.3% | 0.0% | 18.5% | 0.0% |
| OmniAgent | GPT-4o | 0.0% | 27.3% | 33.3% | 27.3% | 30.3% | 8.3% | 14.8% | 0.0% |
| Agent S | GPT-4o | 0.0% | 13.3% | 45.8% | 29.2% | 19.1% | **22.2%** | 22.2% | 0.0% |
| Operator | computer-use | **7%** | 26.7% | 29.2% | 29.2% | 28.6% | 8.3% | 22.2% | 4.5% |
| UFO$^2$-base | GPT-4o | 2.3% | 36.7% | 29.2% | 41.7% | 33.3% | 0.0% | 22.2% | **9.1%** |
| UFO$^2$-base | o1 | 2.3% | 30.0% | 37.5% | 50.0% | 33.3% | 8.3% | 22.2% | **9.1%** |
| UFO$^2$ | GPT-4o | 4.7% | 30.0% | 41.7% | **58.3%** | 33.3% | 8.3% | 44.4% | **9.1%** |
| UFO$^2$ | o1 | 4.7% | **40.0%** | **45.8%** | 50.0% | **38.1%** | 16.7% | **51.9%** | **9.1%** |

**Performance Breakdown.** Table 2 presents a detailed breakdown of success rates (SR) by application type on the WAA and OSWorld-W benchmarks, enabling deeper understanding of where UFO$^2$ achieves particularly strong results and identifying areas for further system-level improvements. Across multiple categories, UFO$^2$ consistently demonstrates superior performance compared to baseline CUAs, particularly in application scenarios demanding deeper OS integration or sophisticated multi-step task execution.

Notably, UFO$^2$ excels in tasks involving web browsers and coding environments. For instance, the strongest configuration (UFO$^2$ with o1) attains an impressive 40.0% SR for web browser tasks—markedly outperforming the next-best baseline (OmniAgent) by over 12%. Similarly, in coding-related workflows, UFO$^2$ (GPT-4o) achieves the highest SR of 58.3%, significantly exceeding all competing CUAs. These results underscore the effectiveness of UFO$^2$'s hybrid GUI-API approach and continuous knowledge integration, which enable more precise action inference, reduce brittleness due to GUI changes, and substantially elevate reliability in multi-step workflows.

The breakdown further reveals a clear correlation between application complexity, popularity, and system-level support. Tasks involving LibreOffice (in the Office category of WAA) uniformly yield lower SRs across all evaluated CUAs, largely due to inadequate adherence to accessibility standards and incomplete UIA support. Conversely, OSWorld-W tasks predominantly utilize Microsoft 365 Office applications, which offer richer OS-native APIs and structured accessibility data, resulting in improved SRs (up to 51.9% for UFO$^2$-o1). This discrepancy highlights the critical role that robust OS-level integration and API availability play in achieving high-quality desktop automation.

Cross-application tasks, especially prominent in OSWorld-W, present an even greater challenge. Such tasks inherently require sophisticated task decomposition and robust inter-agent coordination, pushing CUAs—and even human users—to their limits. Here, UFO$^2$'s multiagent architecture, led by the centralized HOSTAGENT and specialized APPAGENTs, demonstrates notable promise, outperforming other baselines with a 9.1% SR. Although performance remains relatively modest, it clearly illustrates the strength of systematic multiagent collaboration and centralized orchestration in addressing complex scenarios that cross traditional application boundaries.

Overall, these detailed breakdown results validate the system-level design principles of UFO$^2$, particularly its emphasis on deep OS and application-specific integration, multiagent coordination, and flexible action orchestration. While there remains significant potential for further enhancements in niche or less-supported application domains (*e.g.*, custom or legacy software with limited API availability), UFO$^2$'s current architecture already provides a substantial, measurable improvement in practical, real-world desktop automation tasks.

**Error Analysis.** To systematically understand the limitations of UFO$^2$ and identify opportunities for further improvement, we conducted a detailed manual review of all failure cases for UFO$^2$-base (GPT-4o) on both benchmarks. Following a classification framework similar to Agashe *et al.*, Agashe et al., each failure was categorized into one of three distinct system-level categories:

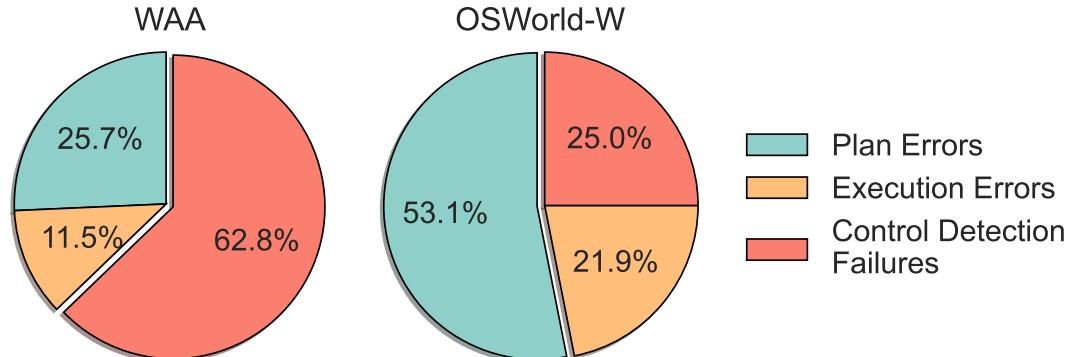

Figure 19: Error analysis of UFO$^2$-base (GPT-4o) on the two benchmarks.

- **Plan Errors:** Failures arising from inadequate high-level task understanding, typically reflected by incomplete or incorrect action plans. These errors indicate gaps in the agent's task comprehension or insufficient grounding in application-specific workflows.

- **Execution Errors:** Cases where the high-level plan is reasonable but the execution is flawed (*e.g.*, selecting an incorrect control, performing unintended actions). Execution errors often stem from inaccurate visual reasoning, incorrect associations between GUI elements and actions, or erroneous inference by the LLM.

- **Control Detection Failures:** Instances where the agent fails to detect or identify critical GUI controls required to complete a task, usually due to non-standard or custom-rendered UI elements that are not fully accessible via standard OS APIs.

Figure 19 summarizes our findings for UFO$^2$-base. On the WAA benchmark, more than 62% of failures were attributed to *Control Detection Failures*, highlighting significant gaps in standard UIA API coverage—especially for third-party applications (*e.g.*, LibreOffice) that do not strictly adhere to accessibility standards. Conversely, the OSWorld-W benchmark exhibited a higher incidence of *Plan Errors*, underscoring that tasks in this set frequently involve more complex workflows, necessitating deeper domain knowledge or advanced contextual reasoning capabilities beyond simple visual recognition.

These observations provide concrete evidence of specific system-level shortcomings, directly motivating the enhancements incorporated into the complete version of UFO$^2$. The high frequency of *Control Detection Failures* validates our choice of adopting a hybrid GUI detection pipeline that supplements standard UIA data with advanced visual grounding techniques. Similarly, the prevalence of *Plan Errors* underscores the critical role of integrating richer external documentation, domain-specific knowledge bases, and application-level APIs to strengthen task understanding and action inference. In the subsequent sections, we explicitly demonstrate how these incremental system-level improvements progressively mitigate each identified category of errors, thereby substantially boosting UFO$^2$'s overall task completion effectiveness.

### 6.3 Evaluation on Hybrid Control Detection

As shown in Figure 19, a considerable fraction of failures arise from *Control Detection Failures*, where non-standard UI elements do not comply with UIA guidelines. To quantify the effectiveness of different detection strategies, we compare UIA-only, OmniParser-v2–only, and our hybrid method (Section 3.4). We introduce a *Control Recovery Ratio (CRR)* to measure how many UIA-only failures are "recovered" (*i.e.*, become successful completions) under OmniParser or the hybrid approach.

Table 3 presents the results on both benchmarks, across multiple model configurations. The hybrid method consistently outperforms either UIA-only or OmniParser-only settings, raising the overall success rate and converting up to 9.86% of previously irrecoverable cases into completions. This gain highlights the complementary strengths of the two detection pipelines, as the hybrid approach bridges coverage gaps in UIA while avoiding OmniParser's limitations in more standardized GUIs.

Table 3: Comparison of SR and CRR across control detection mechanisms.

| Control Detector | Model | WAA | | OSWorld-W | |
|---|---|---|---|---|---|
| | | SR | CRR | SR | CRR |
| UIA | GPT-4o | 23.4% | - | 22.4% | - |
| OmniParser-v2 | GPT-4o | 26.6% | 7.0% | 14.3% | 0% |
| **Hybrid** | **GPT-4o** | **26.6%** | **9.9%** | **22.4%** | **12.5%** |
| UIA | o1 | 25.3% | - | 24.5% | - |
| OmniParser-v2 | o1 | 20.8% | 7.0% | 14.3% | 0% |
| **Hybrid** | **o1** | **27.9%** | **9.9%** | **28.6%** | **25.0%** |

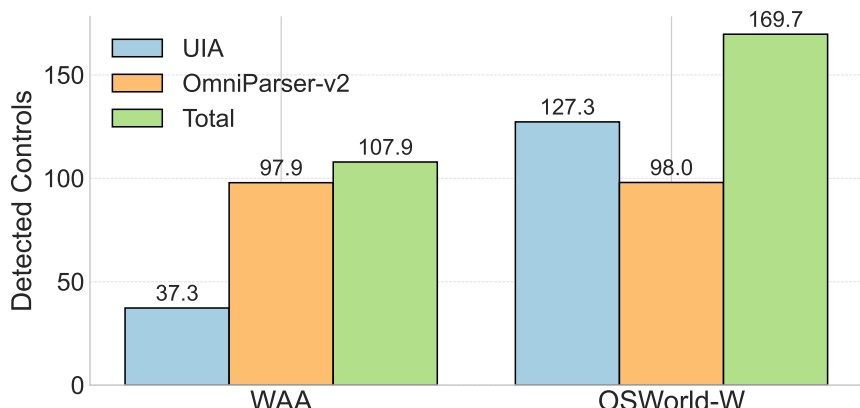

Figure 20: The number of detected controls of different approaches.

In Figure 20, we report the average number of controls detected from each source (UIA, OmniParser-v2, and the merged set) under the hybrid approach. Owing to differences in application coverage, the total number of detected controls is generally higher in OSWorld-W than in WAA. Notably, both UIA and OmniParser-v2 identify substantial subsets of controls, and after merging, 27.9% and 56.7% of OmniParser-v2 detections are discarded due to overlap with UIA. These observations indicate that OmniParser-v2 provides a valuable complement to UIA by recovering nonstandard or custom elements. At the same time, the merging step removes redundancies and prevents double-counting, ultimately reducing control detection failures in the hybrid scheme.

## 6.4 Effectiveness of GUI + API Integration

We now evaluate how unifying API-based actions with standard GUI interactions in the `Puppeteer` impacts performance (Section 3.5). To do so, we focus on the *27* office-related tasks in OSWorld and manually develop *12* APIs for Word, Excel, and PowerPoint. These applications provide COM interfaces that facilitate the creation of custom functions, making them ideal exemplars for deeper OS- and application-level integration. Importantly, many of these operations would require cumbersome multi-step GUI procedures but become straightforward single calls via these APIs (*e.g.*, select paragraphs). Table 4 details the implemented APIs.

Table 5 compares *(i)* overall Success Rate (SR), *(ii)* Plan Error Recovery rate (PRR), *(iii)* Execution Error Recovery rate (ERR), *(iv)* Control Detection Failure Recovery rate (CRR), and *(v)* Average Completion Steps (ACS) for two configurations: GUI-only versus GUI + API. We calculate ACS on the subset of tasks that both configurations successfully complete, ensuring a fair comparison.

The results show that integrating API actions boosts SR for both GPT-4o (+6.1%) and o1 (+8.2%), underscoring the effectiveness of mixing GUI and API interactions. Notably, GPT-4o benefits most from APIs in recovering from *Control Detection Failures* by circumventing unannotated GUI elements. In contrast, o1 more frequently addresses *Plan Errors* through API "shortcuts", reflecting the model's stronger reasoning capabilities and preference for concise solutions.

Table 4: APIs supported across Office applications.

| API | Application | Description |
|---|---|---|
| select_text | Word | Select matched text in the document. |
| select_paragraph | Word | Select a paragraph in the document. |
| set_font | Word | Set the font size and style of selected text. |
| save_as | Word | Save the current document to a desired format. |
| insert_excel_table | Excel | Insert a table at the desired position. |
| select_table_range | Excel | Select a range within a table. |
| reorder_column | Excel | Reorder columns of a table. |
| save_as | Excel | Save the current sheet to a desired format. |
| set_background_color | PowerPoint | Set the background color of slide(s). |
| save_as | PowerPoint | Save the current presentation to a desired format. |

Table 5: Performance comparison of GUI-only vs. GUI + API actions.

| Action | Model | SR | PRR | ERR | CRR | ACS |
|---|---|---|---|---|---|---|
| GUI-only | GPT-4o | 16.3% | - | - | - | 13.8 |
| **GUI+API** | **GPT-4o** | **22.4%** | **5.9%** | **14.3%** | **25.0%** | **12.9** |
| GUI-only | o1 | 16.3% | - | - | - | 16.0 |
| **GUI+API** | **o1** | **24.5%** | **17.7%** | **0.0%** | **12.5%** | **6.6** |

Beyond higher success rates, GUI + API also reduces the effort required to complete tasks. UFO$^2$ achieves a 6.5% step savings with GPT-4o and an impressive 58.5% reduction for o1 on identical tasks. The latter improvement stems from o1's ability to strategically call API functions, bypassing multiple GUI-based steps. Overall, these findings confirm the advantages of mixing GUI automation with API calls, both in terms of robustness and efficiency, and showcase the importance of deep system integration for desktop automation.

**Case Study.** To illustrate how the GUI + API approach streamlines task execution, Figure 21 shows the completion trajectory for exporting an Excel file to CSV format in a case of OSWorld-W, using either GUI-only or GUI + API interactions. Although both configurations eventually succeed, the GUI-only setting requires five steps to open the Save dialog, select the file format, and confirm the action. In contrast, a single call to the save_as API completes the task immediately. Beyond improving efficiency, this one-step solution also reduces the risk of compounding errors across multiple GUI interactions—a clear demonstration of the advantages of deeper OS and application-level integration.

## 6.5 Continuous Knowledge Integration Evaluation

We next evaluate the impact of continuous knowledge integration (Section 3.6) on UFO$^2$'s performance. Specifically, we augment UFO$^2$ with external documentation and execution-derived insights to dynamically improve its domain understanding without retraining. We create *34* help documents tailored to benchmark tasks, each containing precise step-by-step instructions, enabling UFO$^2$ to retrieve the most relevant guidance

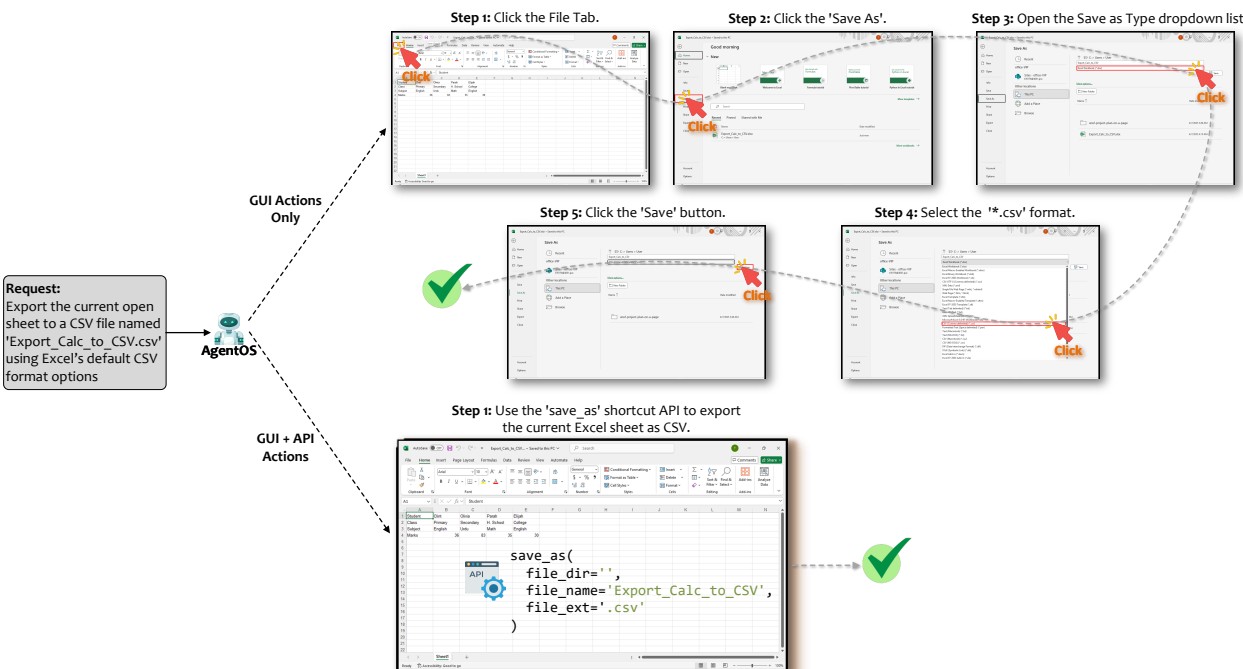

Figure 21: A case study comparing the completion of the same task using GUI-only actions vs. GUI + API actions.

Table 6: Performance comparison with and without knowledge integration.

| Knowledge Enhancement | Model | WAA | | OSWorld-W | |
|---|---|---|---|---|---|
| | | SR | PRR | SR | PRR |
| None | GPT-4o | 23.4% | - | 22.4% | - |
| Help Document | GPT-4o | 26.6% | 10.34% | 26.5% | 11.8% |
| Self-Experience | GPT-4o | 26.6% | 13.79% | 24.5% | 11.8% |
| None | o1 | 25.3% | - | 24.5% | - |
| Help Document | o1 | 27.9% | 3.5% | 28.5% | 17.7% |
| Self-Experience | o1 | 20.8% | 13.79% | 26.5% | 17.7% |

(maximum of one per task) at runtime. Additionally, we implement an automated pipeline that summarizes successful execution trajectories—validated by our Task Evaluator and archives them into a retrievable knowledge database. For subsequent tasks, UFO$^2$ dynamically retrieves up to three relevant past execution logs to guide task planning and execution. Given that knowledge integration primarily addresses failures arising from insufficient planning (*Plan Errors*), we employ the *Plan Recovery Ratio (PRR)* to measure the proportion of previously failed planning cases successfully resolved by integrating new knowledge.

Table 6 compares the overall SR and PRRs across two benchmarks, highlighting significant performance improvements attributable to knowledge integration. Both live help-document retrieval and self-experience summarization yield noticeable gains, reducing planning failures by up to 17.7%. Notably, self-experience enhancements using the stronger model (o1) achieve consistent improvements across both benchmarks, underscoring the efficacy of leveraging prior successes for adaptive improvement. While help documents occasionally result in modest gains, their effectiveness depends on task complexity and document specificity.

These findings underscore the value of systematic knowledge integration, demonstrating that continuous augmentation of the agent's knowledge base can substantially enhance its robustness, scalability, and adaptability in real-world deployments. Moreover, as UFO$^2$ continues to accumulate execution experience and

Table 7: The SR and ACS comparison between single action and speculative multi-action mode.

| Action Execution | Model | WAA | | | OSWorld-W | | |
|---|---|---|---|---|---|---|---|
| | | SR | ACS | Success Subset | SR | ACS | Success Subset |
| Single | GPT-4o | 23.4% | 10.00 | 30 | 22.4% | 13.30 | 10 |
| Speculative | GPT-4o | 23.4% | **8.78** | | 24.5% | **7.40** | |
| Single | o1 | 25.3% | 9.95 | 32 | 24.5% | 6.80 | 10 |
| Speculative | o1 | 24.7% | **8.85** | | 26.5% | **3.30** | |

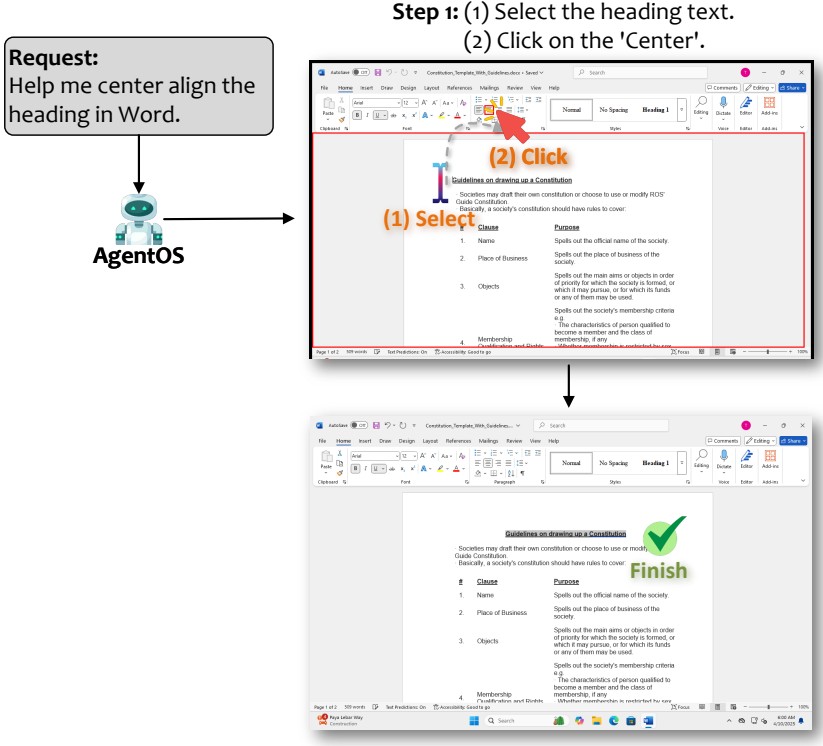

Figure 22: A case study of the successful speculative multi-action execution.

documentation over time, it inherently evolves toward higher reliability and improved autonomy, marking a clear path for ongoing enhancement in desktop automation.

## 6.6 Effectiveness of Speculative Multi-Action Execution

Next, we evaluate how speculative multi-action execution (Section 3.7) affects task completion rates and efficiency. Table 7 compares two modes for UFO$^2$: generating and executing one action per inference (*single-action*) versus inferring multiple consecutive actions in one step (*speculative multi-action*). To ensure a fair comparison, we compute the Average Completion Steps (ACS) only on the subset of tasks that both modes complete successfully.

The results show that speculative multi-action execution retains a comparable Success Rate (SR) to single-action mode while notably cutting the average steps—by up to 10% on WAA and an impressive 51.5% on OSWorld-W. Because each step requires an LLM call, reducing the number of steps significantly lowers both latency and cost. This finding confirms that speculative multi-action planning enhances efficiency without compromising reliability, further highlighting UFO$^2$'s ability to optimize resource utilization in practical desktop automation.

Table 8: Ablation study on the effect of HOSTAGENT orchestration and the shared blackboard.

| Architecture | Model | WAA | OSWorld-W |
|---|---|---|---|
| UFO$^2$ | GPT-4o | 23.4% | 22.4% |
| w/o HostAgent | GPT-4o | 14.3% | 16.3% |
| w/o Blackboard | GPT-4o | 18.2% | 20.4% |
| UFO$^2$ | o1 | 25.3% | 24.5% |
| w/o HostAgent | o1 | 18.2% | 18.4% |
| w/o Blackboard | o1 | 19.5% | 20.4% |

**Case Study.** Figure 22 illustrates how speculative multi-action execution operates in practice. When the user requests that UFO$^2$ center-align a heading in a Word document, the sequence of steps would typically require selecting the heading text and then clicking the `Center` icon. These actions that are sequentially dependent but do not interfere with each other. Instead of treating each action as a separate LLM inference, UFO$^2$ predicts both actions in a single step, leveraging speculative multi-action planning. Consequently, it completes the task with just one LLM call, significantly enhancing efficiency while maintaining accuracy.

### 6.7 The Effect of HostAgent Orchestration and Blackboard

To further examine the contribution of UFO$^2$'s multiagent orchestration mechanism, we conduct an ablation study on two core architectural components: the HOSTAGENT and the shared blackboard. We compare the full UFO$^2$ architecture with two ablated variants. In the *w/o HostAgent* setting, the system disables centralized task decomposition, application selection, and lifecycle management. The user request is directly handled by the corresponding execution agent without high-level orchestration. In the *w/o Blackboard* setting, the HOSTAGENT remains active, but agents cannot exchange structured intermediate states through the shared blackboard; instead, they only rely on local execution context and minimal direct messages. This setting tests whether the blackboard contributes beyond the high-level planning capability of the HOSTA-GENT.

Table 8 shows that both architectural components contribute substantially to task success. Removing the HOSTAGENT leads to the largest degradation. With GPT-4o, the success rate drops from 23.4% to 14.3% on WAA and from 22.4% to 16.3% on OSWorld-W. With o1, the success rate similarly drops from 25.3% to 18.2% on WAA and from 24.5% to 18.4% on OSWorld-W. This confirms that centralized orchestration is not merely an implementation convenience: decomposing user requests into application-level subtasks, selecting the correct application context, and managing application lifecycles are critical for reliable desktop automation.

Removing the shared blackboard also consistently reduces performance. With GPT-4o, the success rate decreases from 23.4% to 18.2% on WAA and from 22.4% to 20.4% on OSWorld-W. With o1, it decreases from 25.3% to 19.5% on WAA and from 24.5% to 20.4% on OSWorld-W. The degradation is smaller than removing the HOSTAGENT, but remains substantial. This suggests that while high-level task orchestration is the dominant factor, structured state sharing is also important, especially for multi-step and cross-application workflows where intermediate artifacts, application states, and dependency information must be reliably propagated between agents.

Overall, the ablation validates the core design choice of UFO$^2$: robust desktop automation benefits from the synergy between a centralized control plane and application-specialized execution agents connected through a shared state substrate. The HOSTAGENT improves task-level planning and coordination, while the blackboard improves consistency, traceability, and inter-agent handoff during execution.

### 6.8 Operator as an AppAgent

To demonstrate the "Everything-as-an-APPAGENT" capability (Section 5.3), we conducted an experiment where UFO$^2$'s HOSTAGENT orchestrator uses *only* Operator as the APPAGENT. In other words, all native APPAGENTs were disabled, leaving Operator to accept subtasks and communicate via the standard UFO$^2$

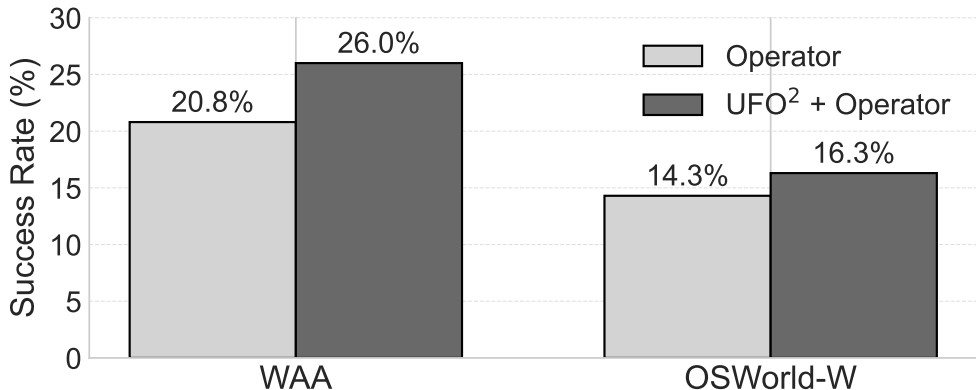

Figure 23: Comparison of Operator vs. $UFO^2$ + Operator on WAA and OSWorld-W.

Table 9: Step count statistic for $UFO^2$.

| Agent | Model | WAA | | | | OSWorld-W | | | |
|---|---|---|---|---|---|---|---|---|---|
| | | HOSTAGENT | APPAGENT | Total | Success Subset | HOSTAGENT | APPAGENT | Total | Success Subset |
| $UFO^2$-base | GPT-4o | **2.21** | 8.11 | 10.32 | 31 | **1.80** | 10.80 | 12.60 | 7 |
| $UFO^2$ | GPT-4o | 2.32 | **7.89** | **10.21** | | 2.80 | **7.20** | **10.00** | |
| $UFO^2$-base | o1 | 2.14 | 7.00 | 9.14 | 34 | 2.50 | 8.83 | 11.33 | 8 |
| $UFO^2$ | o1 | **2.00** | **4.05** | **6.05** | | **2.00** | **3.50** | **5.50** | |

messaging protocol. The only adjustment to Operator's perception layer was restricting it to screenshots of the selected application window, rather than the full desktop.

Figure 23 shows that $UFO^2$ + Operator achieves higher success rates than running Operator alone, particularly on WAA (26.0% vs. 20.8%). We attribute these gains to three key factors. First, the HOSTAGENT breaks down complex user instructions into clearer, single-application subtasks, reducing ambiguity. Second, HOSTAGENT messages include additional tips that improve Operator's decision making. Finally, limiting Operator's view to a single, active application window reduces visual noise and simplifies control detection. Taken together, these results underscore the benefits of $UFO^2$'s multiagent design, while demonstrating how "Everything-as-an-APPAGENT" can elevate the performance of an existing CUA.

### 6.9 Efficiency Analysis

To comprehensively understand the performance characteristics of $UFO^2$, we conducted detailed profiling of task execution efficiency, focusing specifically on step count and latency.

**Step Count Profiling.** Table 9 summarizes the average number of execution steps performed by the HOSTAGENT and APPAGENTs across both benchmark suites. The steps reported are computed on tasks that were successfully completed across all model configurations, ensuring fair comparisons. Two key insights emerge:

First, the fully integrated $UFO^2$ configuration consistently reduces the average number of steps required compared to the baseline ($UFO^2$-base), achieving reductions of up to 50%. This substantial efficiency gain demonstrates how deep OS integration, specifically the hybrid GUI–API action orchestration and advanced control detection strategies, significantly streamline execution paths.

Second, utilizing a more powerful reasoning model (*e.g.*,, *o1* versus GPT-4o) further reduces step counts, indicating that enhanced reasoning capability enables the agent to identify and exploit more efficient action sequences. For instance, stronger models can better leverage direct API interactions or avoid unnecessary intermediate GUI interactions. This underscores the complementary role of both robust system integration and advanced LLM reasoning in minimizing execution overhead.

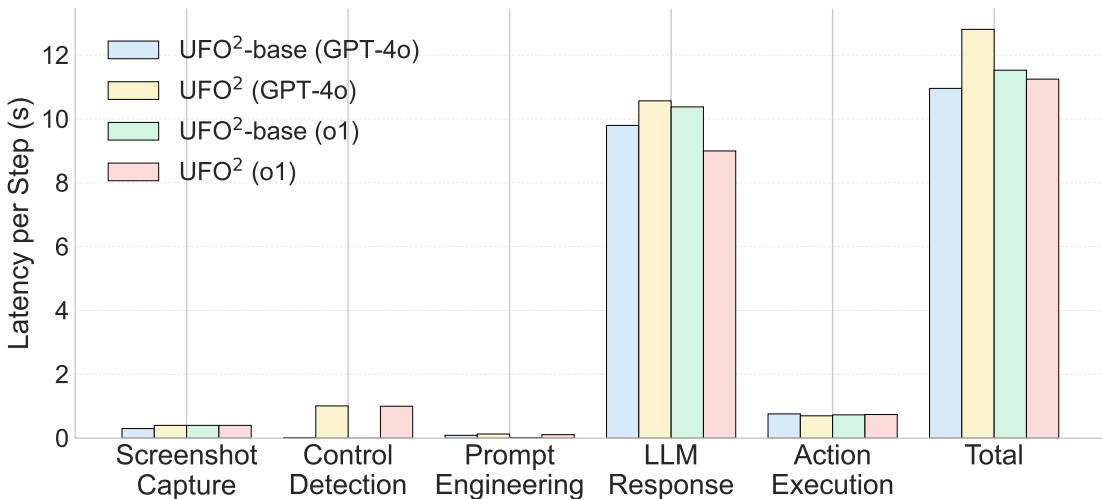

Figure 24: Average time cost per-stage of a single execution step.

**Latency Breakdown.** Figure 24 provides a detailed breakdown of average latency per execution step in UFO$^2$, separated into five key phases: *(i) Screenshot Capture*, *(ii) Control Detection* via UIA APIs (and) OmniParser-v2, *(iii) Prompt Preparation*, including retrieval of relevant help documents and historical execution experiences, *(iv) LLM Inference*, and *(v) Action Execution* on target applications.

Across all configurations, the LLM inference phase dominates total latency, averaging around 10 seconds per inference. This bottleneck highlights a clear opportunity for optimization by deploying smaller, specialized models or employing more powerful inference hardware—strategies that remain viable but are beyond our current evaluation scope.

Excluding LLM inference overhead, the baseline system (UFO$^2$-base) achieves highly efficient execution, incurring around 10 seconds per step on average. In contrast, the fully integrated UFO$^2$ incurs only an additional 1 second per step for its hybrid control detection pipeline, largely due to OmniParser-v2 visual parsing. This added overhead represents a deliberate trade-off, significantly enhancing the robustness and accuracy of GUI control detection at a modest latency cost.

Taken together, these results indicate that the substantial reduction in total steps required per task ensures overall task completion times remain practical (approximately 1 minute per task). These profiling insights reinforce that UFO$^2$'s comprehensive system-level integration balances latency, accuracy, and efficiency, offering a scalable and performant solution for real-world desktop automation.

### 6.10 Model Ablation

Figure 25 compares the performance of UFO$^2$ and UFO$^2$-base across five large language models. GPT-4V and GPT-4o generate direct answers without exposing an explicit CoT, whereas Gemini 2.0 (Flash Thinking) and o1 embed reasoning steps internally before producing final outputs. Overall, models with built-in reasoning typically achieve higher success rates (SR), highlighting the value of more deliberative or CoT-driven processes in desktop automation Lu et al. (2025). In addition, we include Qwen-3.5 Team (2026) as a non-OpenAI model family to examine whether the benefits of UFO$^2$ are tied to a specific model provider. Qwen-3.5 achieves competitive results under the UFO$^2$-base setting and is further improved by the full UFO$^2$ system, indicating that UFO$^2$'s system-level design can benefit models beyond the OpenAI family.

The Qwen-3.5 results also help distinguish model-agnostic system design from model-independent performance. UFO$^2$ is model-agnostic at the interface level: the HostAgent and AppAgents provide structured observations and expect structured action outputs, without depending on model-specific APIs. However, the final success rate still depends on each model's reasoning, visual grounding, and tool-use capabilities. In

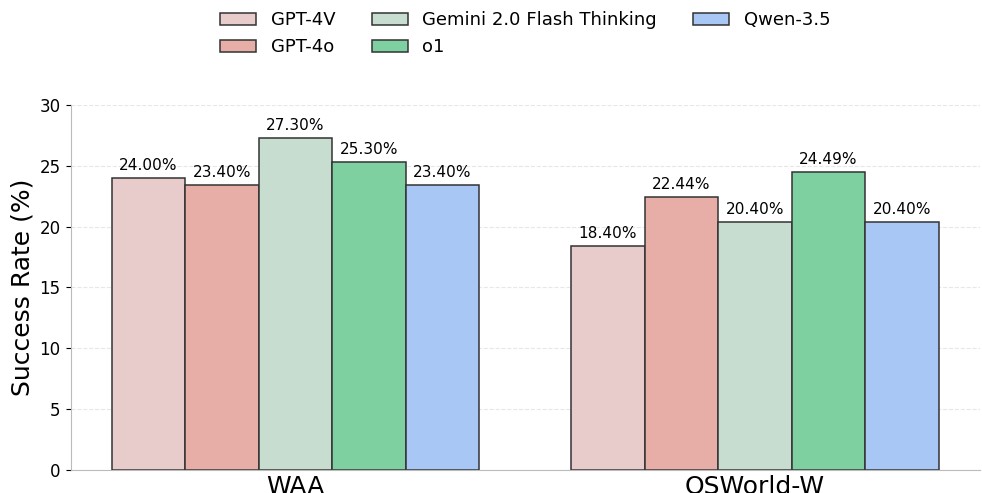

Figure 25: Comparison of different LLMs used in UFO$^2$ and UFO$^2$-base on WAA and OSWorld-W.

our results, stronger reasoning models such as o1 still achieve the best overall performance, while Qwen-3.5 demonstrates that the same AgentOS substrate remains usable and beneficial with a different model family.

This result underscores a promising direction for CUAs: fine-tuning advanced reasoning models specifically for desktop automation tasks. By allowing agents to formulate and refine multi-step plans—especially when integrated with deeper OS-level signals—UFO$^2$ can address complex or ambiguous situations more reliably. As LLM-based reasoning continues to mature, we expect further gains in both accuracy and generality from UFO$^2$'s model-agnostic design.

## 7 Discussion & Future Work

**Latency and Responsiveness.** UFO$^2$ currently invokes LLM inference at each decision step, incurring a latency typically ranging from several seconds to tens of seconds per action. Despite various engineering optimizations, complex tasks comprising multiple sequential actions can accumulate to execution times of 1–2 minutes, which remains acceptable but still inferior to skilled human performance. To alleviate user-perceived delay, we introduced the Picture-in-Picture (PiP) interface, enabling UFO$^2$ to execute tasks unobtrusively within an isolated virtual desktop, thus substantially reducing user inconvenience during longer-running automations. In future work, we aim to further lower latency by investigating the deployment of specialized, lightweight Large Action Models (LAMs) Wang et al. (2024b), optimized for task-specific inference to enhance both responsiveness and scalability.

**Closing the Gap with Human-Level Performance.** Our comprehensive evaluations indicate that UFO$^2$, while robust and effective, does not yet consistently achieve human-level performance across all Windows applications. Bridging this gap will necessitate advances primarily along two critical dimensions. First, enhancing foundational visual-language models through fine-tuning on extensive, diverse GUI interaction datasets will significantly improve agents' capabilities and generalization across varied applications. Second, tighter integration with OS-level APIs, native application interfaces, and comprehensive, structured documentation sources will deepen contextual understanding and bolster execution reliability. Given UFO$^2$'s modular architecture, these enhancements can be incrementally adopted, continuously refining performance towards human-equivalent proficiency across diverse application scenarios.

**Cross-OS Portability and Limitations.** UFO$^2$ is implemented and evaluated as a Windows desktop AgentOS. Therefore, all empirical claims in this paper should be interpreted within the Windows ecosystem, including Windows UI Automation (UIA), COM-based application APIs, Windows Remote Desktop loopback, and Windows-specific process/window management. While the architectural principles of UFO$^2$ are

Table 10: Cross-OS portability analysis of UFO$^2$. UFO$^2$'s agent abstractions are architecturally portable, while the concrete OS backends require platform-specific engineering.

| UFO$^2$ Component | Windows Backend | Possible Linux/macOS Analog | Key Adaptation Challenge |
|---|---|---|---|
| Control introspection | UI Automation (UIA) | AT-SPI; macOS Accessibility API | Different metadata coverage, latency, and permission models |
| Application APIs | COM; app-specific MCP servers | AppleScript/JXA/Shortcuts; DBus; CLI; app SDKs | API availability and semantic equivalence vary across applications |
| Isolated execution | RDP loopback-based PiP | VM/VNC sessions; compositor isolation; sandboxed sessions | User identity, input isolation, and security policies differ by OS |
| Action execution | GUI MCP server + native APIs | Platform-specific GUI automation and tool servers | Reliable coordinate/action mapping across display servers |
| Agent orchestration | HostAgent, AppAgents, blackboard | Same high-level abstraction | Requires OS-specific perception and action backends |

portable, a full cross-platform implementation would require substantial OS-specific engineering rather than a direct reuse of the current Windows backend.

At the architectural level, UFO$^2$ separates the OS-agnostic agent abstractions from OS-specific execution backends. The HostAgent–AppAgent decomposition, shared blackboard, task-level state machines, retrieval-augmented application knowledge, and MCP-based tool registration are not tied to Windows. However, the concrete perception, action, isolation, and permission mechanisms must be re-implemented for each target OS, as shown in Table 10. For example, Windows UIA could be mapped to AT-SPI on Linux or the Accessibility API on macOS, but these frameworks differ in coverage, latency, permission requirements, and metadata quality. Similarly, Windows COM and application-specific APIs would need to be replaced by platform-specific mechanisms such as AppleScript, JXA, Shortcuts, DBus, command-line interfaces, or application-native SDKs. The Picture-in-Picture interface in UFO$^2$ relies on Windows RDP loopback; comparable isolation on Linux or macOS may require VM-based execution, VNC sessions, compositor-level isolation, or platform-specific sandboxing, each with different security and usability trade-offs.

Thus, our claim is not that UFO$^2$ already provides a fully cross-platform AgentOS, but that its layered design exposes a portable blueprint for OS-integrated desktop agents. Building and evaluating Linux and macOS backends is an important direction for future work.

**Safeguard Scope and Limitations.** The safeguard mechanism in UFO$^2$ is designed as a human-in-the-loop risk control layer rather than a complete security solution. By default, actions that match configurable risky-action policies, such as deleting files, overwriting documents, closing applications with unsaved changes, sending external messages, or accessing sensitive devices, trigger the AppAgent's PENDING state and require explicit user confirmation before execution. This design reduces the likelihood of accidental harmful actions and gives users or administrators a simple way to customize risk policies.

Nevertheless, the current safeguard does not eliminate all security risks. Its effectiveness depends on the coverage of the risk policy, the accuracy of the model's action interpretation, and the trust boundary of registered tools and APIs. A malicious or overly permissive tool server could still expose dangerous capabilities if it is registered without proper administrative control. Therefore, practical deployment should combine UFO$^2$'s confirmation mechanism with least-privilege tool registration, organization-level policy enforcement, audit logging, and explicit approval for destructive or externally visible actions. A comprehensive adversarial evaluation of safeguard false positives and false negatives is left for future work.

# 8 Related Work

Integrating LLMs into OS represents a growing, yet nascent area of research. In this section, we discuss prior work that intersects with our research on system-level integration of multimodal LLM-based desktop automation agents.

## 8.1 Computer-Using Agents (CUAs)

Recent advancements in multimodal LLMs have significantly accelerated the development of *Computer-Using Agents* (CUAs), which automate desktop workflows by simulating GUI interactions at the OS level. Early pioneering systems, such as UFO Zhang et al. (2024b), leveraged multimodal models (*e.g.*, GPT-4V Yang et al. (2023b)) alongside UIA APIs to interpret graphical interfaces and execute complex tasks via natural language instructions. UFO notably introduced a multi-agent architecture, enhancing the reliability and capability of CUAs to handle cross-application and long-term workflows.

Subsequent efforts have focused primarily on refining underlying multimodal models and extending platform capabilities. For example, CogAgent Hong et al. (2024), built upon the vision-language model CogVLM Wang et al. (2024c), specialized in GUI understanding across multiple platforms (PC, web, Android), representing one of the earliest dedicated multimodal CUAs. Industry interest has similarly accelerated with Anthropic's Claude-3.5 (Computer Use) Anthropic (2024), an agent relying entirely on screenshot-based GUI interactions, and OpenAI's Operator OpenAI (2025), which significantly improved desktop automation performance through advanced multimodal reasoning.

However, these existing CUAs remain largely prototype demonstrations, often lacking deep integration with the OS and native application capabilities. In contrast, our work in UFO$^2$ directly addresses these fundamental system-level limitations through a modular AgentOS architecture, deep OS and API integration, hybrid GUI detection, and a non-disruptive execution model, bridging the gap between conceptual CUAs and practical desktop automation.

## 8.2 LLMs for Operating Systems

Another promising research direction involves embedding LLMs directly within OS architectures, aiming to substantially enhance automation, adaptability, and usability. Ge *et al.*, first proposed the conceptual framework AIOS Ge et al. (2023), positioning an LLM at the center of OS design to orchestrate high-level user interactions and automated decision-making. In their vision, agents resemble OS applications, each exposing specialized capabilities accessible via natural language, effectively enabling users to "program" their OS intuitively.

Building on this conceptual foundation, Mei *et al.*, Mei et al. (2024) realized AIOS as a concrete prototype, encapsulating LLM interactions and tool APIs within a privileged OS kernel. This design provides core OS functionalities such as process scheduling, memory management, I/O handling, and access control, leveraging LLMs to simplify agent development through a dedicated SDK. Rama *et al.*, Rama et al. (2025) extended this paradigm, introducing semantic file management capabilities directly within traditional OS environments through AIOS-based agents, further demonstrating practical system-level integration.

Complementing these high-level OS integrations, AutoOS Chen et al. (2024b) applied LLMs for automatic tuning of kernel-level parameters in Linux, achieving substantial efficiency gains through autonomous exploration and optimization. This highlights another dimension where LLM integration can directly enhance core system performance and management.

Collectively, these research efforts illustrate an emerging paradigm shift where LLMs become integral components of operating systems, enabling powerful automation, enhanced user interaction, and adaptive system behavior. Our work with UFO$^2$ extends this line of research specifically to desktop automation, offering a deeply integrated, scalable, and practical AgentOS that leverages multimodal LLMs in conjunction with robust OS-level mechanisms.

## 9 Broader Impact

UFO$^2$ aims to make desktop automation more robust, accessible, and less disruptive by integrating computer-using agents with OS-level perception, application APIs, and isolated execution environments. Potential benefits include reducing repetitive office work, improving accessibility for users who have difficulty operating complex GUIs, and enabling more reliable automation of multi-application workflows. By allowing users to supervise long-running tasks through the PiP interface and confirmation prompts, UFO$^2$ may also lower the barrier to using desktop automation in practical settings.

At the same time, desktop automation agents introduce important societal and security risks. First, broad automation of office workflows may affect labor demand for repetitive clerical tasks. We believe such systems should be deployed as assistive tools that augment human workers, with clear human oversight and opportunities for task redesign and reskilling. Second, because UFO$^2$ can access application state, files, native APIs, and user-level credentials, it may create risks of unauthorized data access, privacy leakage, or accidental modification of sensitive information. Third, malicious users could attempt to use desktop agents for harmful automation, such as bulk messaging, data exfiltration, or unauthorized manipulation of enterprise systems.

UFO$^2$ includes several mechanisms that partially mitigate these risks, including human confirmation for risky actions, configurable safeguard policies, PiP-based execution isolation, structured logging, replayable traces, and administrator-controlled tool/API registration. However, these mechanisms should not be interpreted as complete security guarantees. Responsible deployment should follow least-privilege principles, restrict available tools to approved capabilities, avoid exposing unnecessary sensitive data to external models, retain audit logs for accountability, and require explicit approval for destructive or externally visible actions. Further research is needed on adversarial safety evaluation, formal permission models for computer-using agents, and enterprise governance mechanisms for agentic desktop automation.

## 10 Conclusion

We introduced **UFO$^2$**, a practical, OS-integrated Windows desktop automation AgentOS that transforms CUAs from conceptual prototypes into robust, user-oriented solutions. Unlike prior CUAs, UFO$^2$ leverages deep system-level integration through a modular, multiagent architecture consisting of a centralized HOSTAGENT and application-specialized APPAGENTs. Each APPAGENT seamlessly combines GUI interactions with native APIs and continually integrates application-specific knowledge, substantially improving reliability and execution efficiency. UFO$^2$ can operate on a PiP virtual desktop interface further enhances usability, enabling concurrent user-agent workflows without interference.

Our comprehensive evaluation across over 20 real-world Windows applications demonstrated that UFO$^2$ achieves significant improvements in robustness, accuracy, and scalability compared to state-of-the-art CUAs. Notably, by coupling our integrated framework with robust OS-level features, even less specialized foundation models (*e.g.*, GPT-4o) surpass specialized CUAs such as Operator.

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
