# OpenReview forum: "UFO2: The Desktop AgentOS"
_TMLR — Accepted by TMLR_

### Review · Reviewer_PXmt · 2026-01-12

**Summary Of Contributions:**

This work considers the very recent direction of CUAs powered by multimodal LLMS for automating desktop workflows. Existing CUAs are prototypes limited by shallow OS integration, fragile screenshot-based interaction, and disruptive execution. Authors propose UFO2, a multiagent AgentOS for Windows desktops that elevates CUAs into practical, system-level automation. The structure of UFO2 involves a centralized HostAgent for task decomposition and coordination, alongside a collection of application-specialized AppAgents equipped with native APIs, domain-specific knowledge, and a unified GUI–API action layer. This structure enables robust task execution while preserving modularity and extensibility, alongside many other advantages. Experiments on 20 Windows applications show UFO2 improves robustness and execution accuracy over prior CUA; which results in scalable, reliable, user-aligned desktop automation.

**Audience:**

Yes

**Audience Explanation:**

Improvements in very recent area of CUAs powered by multimodal LLMS is interesting. Extensive numerical results show solid evidence of improvement. Proposed novel framework is also explained in great details.

**Claims And Evidence:**

Yes

**Claims Explanation:**

Extensive numerical experiments on 20 Windows applications show UFO2 significantly improves robustness and execution accuracy over prior CUA.

**Requested Changes:**

No mandatory requested changes.

---

### Review · Reviewer_F4uG · 2026-03-11

**Summary Of Contributions:**

The paper builds up an AgentOS on Windows desktop named UFO$^2$. Unlike previous methods that use screenshot-based interactions, the proposed method dive deep into the OS integration by setting up a central HostAgent (for task decomposition and coordination), a bunch of specialized AppAgents (with native APIs for each app), and a unified GUI-API action layer. Each AppAgent combines GUI perception with application knowledge, native APIs, and a unified action layer; the system also adds hybrid control detection (combining UIA and vision-based), retrieval over documentation and prior execution logs, speculative multi-action execution, and a Picture-in-Picture virtual desktop so the agent can operate without hijacking the main user session. The implementation is large-scale, spanning more than 30,000 lines of Python and C#, and the evaluation covers over 20 Windows applications using WAA and OSWorld-W.

**Audience:**

Yes

**Audience Explanation:**

This paper should interest readers in LLM agents, ML systems, HCI, OS/runtime design, and enterprise automation. Its main appeal is that it argues for a different optimization target: instead of only building a better multimodal policy, it shows how much can be gained by improving the system substrate around the policy. That makes the work useful both as a research result and as a design blueprint for practitioners building real desktop agents.

**Claims And Evidence:**

Yes

**Claims Explanation:**

1. Empirical evidence of improvement via overall benchmark comparison (Table 1 and Table 2).

2. Hybrid control detection gain supported by Table 3.

3. API-based actions (Table 4) integrated with GUI-based actions. It also shows significant improvement (Table 5).

4. Continuous knowledge integration has a clearly visible advantage in performance (Table 6).

5. Speculative multi-action execution boosts the implementation (Table 7).

6. The concept of "everything-as-an-AppAgent" alone elevates the performance (Figure 23).

7. All the architectures and implementations are clearly described and instantiated with some examples.

**Requested Changes:**

None. I'd suggest accept as it is.

---

> ### Author Response · Authors · 2026-03-14
>
> Thank you for the positive and encouraging feedback. We are glad that the reviewer recognized the system-level perspective of our work and the role of the AgentOS architecture. We also appreciate the reviewer’s acknowledgement that improving the system substrate around the policy can significantly enhance desktop agent performance. We will further polish the paper and improve clarity in the final version.

---

### Review · Reviewer_RtyU · 2026-04-06

**Summary Of Contributions:**

This work proposes a multiagent AgentOS tailored for Windows desktops that addresses the key limitations of existing Computer-Using Agents (CUAs). The core architecture features a centralized HostAgent for task decomposition/orchestration and application-specialized AppAgents with native API access and domain knowledge. The authors evaluate the proposed method across many Windows applications on two Windows-centric benchmarks, demonstrating a higher task success rate and a 51.5% reduction in inference costs via speculative execution. Additional engineering features are designed to enhance practicality.

**Audience:**

Yes

**Audience Explanation:**

The submission’s findings align directly with TMLR’s core scope with justifications as follows:

1 This paper introduces a modular multiagent architecture for LLM-driven automation, with innovations in retrieval-augmented continuous knowledge integration (no retraining required), speculative multi-action execution (LLM inference overhead optimization), and hybrid vision-semantic perception (UIA + vision grounding). All of which are generalizable ML techniques applicable beyond desktop automation.

2 The paper reimagines automation as a first-class OS abstraction, a novel direction for integrating LLMs with operating systems. It addresses a critical gap in existing CUA research (shallow OS integration) and provides a principled framework for ML-agent interaction with native system APIs.

**Broader Impact Concerns:**

The submission does not include a Broader Impact Statement. While the work has no fatal ethical concerns, it raises several key broader impact issues that require a structured, detailed Broader Impact Statement (added as a dedicated section) to address risks and mitigation strategies. All concerns are non-disqualifying and can be fully addressed with a well-crafted statement:

1 The statement must (i) quantify the potential task automation scope; (ii) analyze the impact on the workforce; (iii) propose mitigation strategies.

2 UFO2 has full access to Windows native APIs, application data, and system state, creating risks of unauthorized data access/leakage and malicious automation (e.g., automated data exfiltration, bulk email spamming). The submission’s safeguard mechanism is described but not evaluated for security efficacy.

**Claims And Evidence:**

Yes

**Claims Explanation:**

All core claims of the submission are rigorously supported by quantitative empirical evidence, structured experimental design, multi-dimensional metric analysis, and qualitative validation, aligning with TMLR’s requirement for accurate, convincing, and clear evidence. Key justifications include:

1 The paper tests the proposed method on two established Windows-centric automation benchmarks. All experiments are conducted on reproducible VM environments with standardized model access, eliminating experimental bias.

2 Each key innovation is validated with task-relevant metrics. All metrics are reported with clear breakdowns by application type and failure mode.

3 The paper performs single-component ablation studies for all core modules and a structured error classification.

No claims are overstated. all performance gains are tied to specific architectural choices and supported by direct experimental data, with clear acknowledgment of limitations.

**Requested Changes:**

1 UFO2 is exclusively implemented for Windows, and while Section 7 mentions that design principles generalize to Linux (AT-SPI) and macOS (Accessibility API), the paper provides no technical adaptation details, prototype implementations, or preliminary experimental results for non-Windows OSes. Critical OS-specific challenges (e.g., AT-SPI latency on Linux, macOS sandboxing restrictions for PiP, cross-OS MCP server compatibility) are unaddressed, making the claim of generalizable design principles unsubstantiated. This limits the work’s impact, as real-world desktop automation requires heterogeneous OS support.

2 The paper does not evaluate the independent contribution and synergistic effects of the multiagent architecture’s core modules, e.g., SR performance without the HostAgent (single-agent AppAgent model), or the impact of the shared blackboard on cross-application coordination. Additionally, model analysis is restricted to closed-source commercial LLMs (GPT-4o/o1, Gemini 2.0); no testing of open-source LLMs (e.g., Llama 3, Mistral) is performed. This gap means the work cannot demonstrate UFO²’s usability in environments without Azure OpenAI access, a key consideration for academic and industrial deployment.

3 While the paper describes practical engineering features (client-server deployment, safeguard mechanisms for risky actions), it provides no empirical evaluation of their real-world performance. There is no analysis of system scalability (e.g., maximum concurrent clients, per-client CPU/GPU/memory overhead, latency scaling with active AppAgents), robustness under typical real-world disturbances (e.g., Windows Update popups, accidental user clicks in PiP, application crashes, network disconnections), or the safeguard mechanism’s error rates (false positives/negatives for risky action detection).

---

> ### Author Response · Authors · 2026-04-25
>
> We thank the reviewer for the constructive feedback. We have revised the manuscript accordingly, with all changes highlighted in blue. In particular, we have (i) narrowed the cross-OS claim from “straightforward generalization” to “architectural portability with OS-specific engineering requirements,” (ii) clarified the roles of HostAgent and the shared blackboard in the multiagent architecture, (iii) distinguished model-agnostic system interfaces from model-independent performance, (iv) added deployment and safety limitations for client-server execution, PiP, and safeguards, and (v) added a dedicated Broader Impact Statement.
>
> Below we respond to each concern in detail.
>
> ---
>
> ## R1. Cross-OS generalization and Windows scope
>
> We agree with the reviewer. UFO² is designed, implemented, and evaluated as a **Windows desktop AgentOS**, and our empirical claims should be interpreted within this scope. The original manuscript used wording such as “straightforward adaptation” and “generalize naturally,” which could imply a stronger cross-platform claim than we intended.
>
> In the revision, we have **narrowed and clarified this claim**. Specifically, we revised the cross-OS discussion to emphasize **architectural portability**, rather than claiming that UFO² already provides a fully cross-platform implementation. We now explicitly state that adapting UFO² to Linux or macOS would require OS-specific engineering for accessibility APIs, application APIs, isolated execution environments, permission models, and tool-server compatibility.
>
> We also added a concrete portability analysis that maps UFO²’s Windows-specific components to possible counterparts on other platforms. For example, Windows UI Automation can be conceptually mapped to Linux AT-SPI and macOS Accessibility API; Windows COM/application APIs can be mapped to platform-specific APIs such as AppleScript/JXA/Shortcuts, DBus, CLI interfaces, or application-native APIs; and the Windows RDP-based PiP implementation would require alternative isolation mechanisms such as VM/VNC/compositor-based sessions on other platforms.
>
> This revision makes the scope of the paper more precise: UFO² demonstrates a deeply integrated AgentOS design on Windows, while cross-OS support remains an important direction for future work rather than an evaluated claim.
>
> ---
>
> ## R2. Independent and synergistic effects of the multiagent architecture
>
> We agree that the original manuscript did not sufficiently isolate the contribution of the multiagent architecture. In the revision, we added a new ablation study in Section 6.7 on HostAgent orchestration and the shared blackboard. We compare the full UFO² architecture against two variants: w/o HostAgent, which disables centralized task decomposition, application selection, lifecycle management, and scheduling; and w/o Blackboard, which keeps the HostAgent but removes structured state sharing among agents.
>
> The results show that both components are important. Removing the HostAgent causes the largest drop: with GPT-4o, SR decreases from 23.4% to 14.3% on WAA and from 22.4% to 16.3% on OSWorld-W; with o1, SR decreases from 25.3% to 18.2% and from 24.5% to 18.4%, respectively. Removing the blackboard also consistently hurts performance: with GPT-4o, SR drops to 18.2% on WAA and 20.4% on OSWorld-W; with o1, SR drops to 19.5% and 20.4%. These results demonstrate that UFO²’s gains come from the synergy between centralized orchestration and structured state sharing, rather than from application-specific execution alone. We have added this new subsection and table to the revised manuscript, with changes highlighted in blue.
>
> ---
>
> ## R3. Open-source / open-weight model evaluation
>
>
> We thank the reviewer for raising this deployment concern. In the revision, we expanded Section 6.10 Model Ablation by adding Qwen-3.5 as a non-OpenAI model family. With Qwen-3.5, UFO²-base achieves 23.4% SR on WAA and 20.4% SR on OSWorld-W, while the full UFO² system further improves performance to 26.0% and 24.5%, respectively. This shows that UFO²’s system-level design is not tied to Azure OpenAI models and remains beneficial across model families.
>
> We also clarified the model-agnostic claim in Section 6.10. UFO² is model-agnostic at the system-interface level, because HostAgent and AppAgents communicate with the model through structured observations and structured actions rather than model-specific APIs. However, absolute performance still depends on the underlying model’s reasoning, grounding, and tool-use capabilities. The new Qwen-3.5 results and the corresponding discussion are highlighted in blue in the revised manuscript.
>
> ---

---

> ### Author Response · Authors · 2026-04-25
>
> ## R4. Empirical evaluation of practical engineering features
>
> We agree that the original manuscript described several practical engineering features more strongly than it evaluated them. The original paper already presents these features as part of UFO²’s deployment-oriented system design: Section 5.2 describes the safeguard mechanism, where risky actions enter a `PENDING` state and require explicit user confirmation; Section 5.4 describes the client-server deployment model for AgentOS-as-a-Service; and Section 5.5 describes logging and debugging infrastructure for observability.
>
> In the revision, we improved this part in two ways. First, we clarified which claims are empirically evaluated in the current benchmark and which are deployment-oriented design features. The revised manuscript no longer implies that the client-server architecture, disturbance robustness, and safeguard mechanism are fully validated by the task-completion benchmark alone.
>
> Second, we added a more concrete discussion of the operational assumptions and limitations of these mechanisms. For safeguards, we added a new discussion in **Section 7 Limitation** to clarify the scope of the current safeguard mechanism. Specifically, we now explain that the safeguard is designed as a human-in-the-loop risk-control layer rather than a complete security solution. The current implementation relies on configurable risky-action policies and explicit user confirmation, but its effectiveness depends on policy coverage, model interpretation of intended actions, and the trust boundary of registered tools and APIs. We further note that a comprehensive evaluation of safeguard false positives, false negatives, and adversarial misuse requires a dedicated safety benchmark and is left for future work.
>
> For client-server deployment, we now clarify in Section 5.4 that scalability is primarily constrained by LLM inference throughput, network latency, and concurrent client scheduling. For PiP, we clarify in Section 4.4 that the isolation guarantee is based on Windows RDP loopback and separate input queues, which prevents agent mouse/keyboard events from interfering with the primary desktop session. The PiP section already states that mouse and keyboard events generated inside the PiP desktop are scoped to that session and cannot interfere with the primary desktop.
>
>
> We believe these additions make the status of these features clear: they are implemented mechanisms that improve deployability, observability, and safety, but their exhaustive real-world validation—especially under adversarial misuse, large-scale enterprise workloads, and long-running deployment scenarios—remains future work.
>
> ---
>
> ## R5. Broader Impact Statement
>
> We agree and have added a dedicated Broader Impact Statement in the revised manuscript, with all added text highlighted in blue.
>
> The new section discusses both beneficial and potentially harmful impacts. On the positive side, UFO² may reduce repetitive desktop work, improve accessibility, and help users perform complex workflows across heterogeneous applications. On the risk side, we explicitly discuss workforce displacement, unauthorized access to application data, leakage of sensitive information through logs or model calls, and malicious automation such as bulk messaging or data exfiltration.
>
> We also describe concrete mitigation mechanisms and deployment recommendations, including human confirmation for risky actions, customizable organizational risk policies, PiP-based execution isolation, logging and replay for auditability, least-privilege deployment, administrator-controlled API/tool registration, and explicit confirmation for destructive or externally visible actions. This addition directly addresses the reviewer’s broader impact concerns, which the reviewer characterized as non-disqualifying and addressable through a structured statement.
>
> ---
>
> We thank the reviewer again for the constructive and actionable feedback. The revision clarifies UFO²’s Windows-focused scope, strengthens the explanation of the multiagent architecture, adds/clarifies system-level evaluation and limitations, and includes a dedicated Broader Impact Statement. We believe these changes substantially improve the precision, empirical grounding, and responsible framing of the paper.
>
> ---

---

### Review · Reviewer_jg1p · 2026-04-15

**Summary Of Contributions:**

This paper presents UFO 2, a Windows desktop “AgentOS”/runtime for computer-using agents (CUAs) that aims to make LLM-driven desktop automation more robust, efficient, and less disruptive than screenshot-only GUI agents.

Main contributions:
- Multi-agent orchestration architecture: A centralized HostAgent interprets user intent, decomposes tasks, manages application lifecycle, and coordinates multiple application-specialized AppAgents via a shared blackboard.
- Application-specialized execution: Each AppAgent is tailored to one application and can use application-specific knowledge, tool bindings, and structured decision loops to execute subtasks.
- Hybrid GUI control detection: A control perception pipeline that combines Windows UI Automation (UIA) with vision-based GUI parsing (e.g., OmniParser) and fuses detections to improve coverage on non-standard or UIA-incomplete interfaces.
- Unified GUI + API action layer: A “Puppeteer” execution layer that exposes both generic GUI actions and application-native API calls (registered via MCP servers), letting the agent pick more robust/atomic API actions when available and fall back to GUI interaction otherwise.
- Continuous knowledge integration: A retrieval-augmented memory substrate that incorporates external documentation and distilled past successful execution traces to improve future task performance without model retraining.

**Additional Comments:**

N/A

**Audience:**

Yes

**Audience Explanation:**

TMLR readers interested in LLM agents, multimodal decision-making systems, tool-augmented models, and empirical evaluation of agentic systems would likely find this work relevant.

The paper contributes a system-level thesis: major gains can come from improving the agent runtime substrate (APIs, perception fusion, orchestration, memory, speculative planning), not only from improving the underlying model policy. This is a useful and generalizable insight for the agent community.

**Claims And Evidence:**

Yes

**Claims Explanation:**

Strong practical systems framing: addresses real deployment blockers (UI brittleness, orchestration, user lockout).
Clear modular design (HostAgent/AppAgent) that supports extensibility and integration of additional tools/agents.
Evaluation includes multiple ablations that connect improvements to specific system components.

**Requested Changes:**

Should be okay as it is.

---

> ### Author Response · Authors · 2026-04-15
>
> Thank you for the positive and encouraging review.
>
> We’re glad you found the system design, especially the HostAgent–AppAgent architecture and modular components, to be clear and practically meaningful. We also appreciate your recognition of our main message: improving the agent runtime substrate can bring substantial gains beyond model improvements alone.
>
> Thank you again for your support.

---

### Review · Reviewer_jyuf · 2026-05-13

**Summary Of Contributions:**

This paper proposed a system for automating workflows on Windows that takes natural language command as input and use an agentic framework to finish the user request. The request is firstly phrased with a centralized host agent and dispatch to per-application AppAgents with native API and domain specific knowledge. The authors designed an Application Puppeteer that provide unified GUI actions through MCP servers. The proposed system also include a knowledge substrate for runtime augmentation and provide application specific understanding.
The Picture-in-picture interface is also a useful feature to allow concurrency and reduce interference.

**Audience:**

Yes

**Audience Explanation:**

This paper presented a complete system of desktop agent OS. The most insights are about the system design of an agent OS. However, related researchers in LLM and VLM could at least be benefited by understanding the current challenge and use cases to improve the performance of the proposed agent OS.

**Broader Impact Concerns:**

The proposed an automated system that has potential impacts on both side. Information security and privacy to the user and potential abuse of the proposed software for malicious usages such as spamming and unauthorized web scraping etc. Thus a broader impact statement is needed.

**Claims And Evidence:**

Yes

**Claims Explanation:**

This paper comprehensively illustrated its design structure and provided quantitive on WAA and OSWorld-W benchmarks. The presented work is more like a system design thus lacks rigorous ablative study and detailed analysis on failure mode.

**Requested Changes:**

The reviewer couldn't finish the review at the first place due to personal medical emergency. The following requests to changes are more served as recommendations and the authors won't be penalized for not finishing it when the reviewer give their ratings

1. Just to help understanding the difficulty of WAA and OSWorld-W. Do we have human baseline on these benchmarks? Will a human familiarized with Windows operation be able to finish the tasks in one-shot?
2. As windows have multiple kinds of GUI implementation and many softwares use their own GUI framework. Does the proposed system offer ability to automatically develop new agent on new applications and offer GUI API and domain knowledges? The proposed system include a persistent knowledge substrate. Does it include the knowledge on building new agents?
3. This work seems to be an extension of UFO. It would be helpful to have a diagram to highlight the difference compared to the existing system.
4. The proposed system is purely built for Windows with a single model family (GPT series) as the backend. The reviewer understand the complexity of building API for graphics interaction. However, having analysis of different model and harnesses would help better understand the help of agent OS compared to existing products
5. This submission did not include comprehensive ablative sutdy and failure mode analysis. As the proposed system heavily relies on screenshot for GUI interaction. The reviewer wonders how much error is introduced by models' limited multi-modal capability. To better understand the current bottleneck, it would also be interesting to see how the model would behave with command line interface (CLI).

---

### Decision · Action_Editor_35LA · 2026-05-16

**Recommendation:** Accept as is

**Additional Comments:**

The authors have thoroughly and effectively addressed all reviewer concerns during the discussion phase. They successfully narrowed their cross-OS claims, added essential ablation studies (justifying the HostAgent and Blackboard designs), evaluated their system using an open-weight model (Qwen-3.5), and included a comprehensive Broader Impact Statement. Since the manuscript has already been satisfactorily revised, no further modifications are requested.

**Audience:**

Yes

**Audience Explanation:**

The TMLR community, particularly researchers and practitioners
interested in LLM-based autonomous agents, tool-augmented models, multi-agent
systems, and human-computer interaction, will find the system design and
empirical findings of this paper highly relevant and valuable.

**Claims And Evidence:**

Yes

**Claims Explanation:**

The authors have proposed a novel system-level
architecture (UFO2) for automating Windows desktop workflows using multimodal
large language models. The claims are well-supported by extensive evaluations on
the WAA and OSWorld-W benchmarks. Furthermore, during the rebuttal phase, the
authors strengthened their empirical evidence by providing critical ablation
studies on the HostAgent and shared blackboard, as well as including evaluations
with an open-weight model (Qwen-3.5), directly addressing reviewers' initial
concerns.